# VIQS: Overcoming the Teacher Ceiling with Value-Guided Intervention and Quality-Aware Shaping

## Abstract

Human-in-the-loop reinforcement learning (HIL-RL) incorporates real-time human expert intervention and guidance to address the challenges of brittle reward engineering and learning efficiency. However, existing HIL-RL methods primarily rely on direct action mimicry or rigid value alignment, which inherently suffer from a *teacher-quality ceiling*—their performance is fundamentally bounded by the human expert's proficiency due to the absence of mechanisms for assessing guidance quality. To overcome this limitation, we propose a novel framework that integrates two synergistic innovations—**V**alue-guided **I**ntervention and **Q**uality-aware **S**haping (VIQS)—within a reward-free setting. This design allows the agent to break the teacher-quality ceiling by learning robustly from sparse and potentially imperfect expert guidance. First, we propose a *value-guided intervention* mechanism where expert intervention is triggered precisely when the agent's chosen action yields significantly lower estimated long-term value compared to an expert-derived reference, preserving autonomy for strategic exploration. Second, we develop a *quality-aware shaping* mechanism that employs a discriminator to dynamically assess and adaptively incorporate expert intervention data, enabling the agent to filter suboptimal advice while absorbing high-quality guidance. Extensive evaluations are conducted on the challenging MetaDrive benchmark, where pre-trained agents emulate human experts of varying proficiency levels to guide the learning process. Results show that VIQS significantly outperforms prior HIL approaches, while requiring up to $5\times$ fewer interventions. Crucially, it consistently breaks the teacher-quality ceiling across all levels of expert proficiency. Furthermore, integrating our core mechanisms into existing HIL algorithms yields significant and consistent improvements across baselines.

## 1 Introduction

Reinforcement Learning (RL) has delivered remarkable breakthroughs in solving complex sequential decision-making tasks. Nevertheless, a significant challenge in its successful implementation stems from the meticulous design of reward functions, a process known as reward engineering. (Krakovna et al., 2020; Russell, 2022; Leike et al., 2018; Park et al., 2024). Imperfect rewards can lead to suboptimal policies, subtle misalignments with human intent, and even critical safety failures (Hadfield-Menell et al., 2017). Furthermore, this reliance on extensive, trial-and-error interaction is not only prohibitively data-inefficient, particularly for training embodied agents in dynamic environments. These challenges underscore a critical need for learning paradigms that can ensure both efficiency and safety from inception.

A promising paradigm to address these limitations is HIL-RL. By integrating real-time human guidance, HIL-RL offers a potent approach to bypass many reward engineering complexities and significantly enhance learning efficiency (Celemin et al., 2022; Spencer et al., 2020; Wu et al., 2022). Among various HIL approaches, **active intervention frameworks**, where human experts provide real-time guidance by taking over control or offering corrective demonstrations, are particularly effective. This proactive human involvement is crucial for stabilizing learning, curtailing risky exploration, and expediting skill acquisition, especially for embodied agents operating in safety-critical

domains (Kelly et al., 2019; Saunders et al., 2018; Reddy et al., 2018). Our work is situated in this practical and interactive active HIL-RL domain.

Despite its great potential, current HIL-RL methods face a dilemma: **when** human experts should intervene, and the contradiction with **how** effectively the agent learns from these interventions.

On one hand, many approaches still rely on naive **action-based triggers** (Kelly et al., 2019; Li et al., 2022b; Peng et al., 2021; 2025). This design often forces stylistic mimicry and inadvertently caps the agent's performance at the expert's proficiency level—a phenomenon we term the "teacher-quality ceiling". This is particularly problematic in safety-critical domains like autonomous driving, where human performance can be suboptimal or highly variable due to fatigue, workload, or limited situational awareness.

On the other hand, a schism exists in the learning paradigm for processing these interventions. Some methods, while employing sophisticated, value-based triggers to identify strategic inferiority, still **convert the intervention event into an external reward signal** (Xue et al., 2023; Gokmen et al., 2023). This risks signal conflict, reintroduces the very challenge of reward design that HIL-RL aims to mitigate, and can lead to inefficient knowledge retention under sparse guidance (Zuo et al., 2017). Conversely, other methods such as PVP (Peng et al., 2023) and PVP4REAL (Peng et al., 2025), learning without explicit rewards but do so by **blindly trusting the human expert**. They often assign fixed proxy values, overlooking the inherent imperfection and variability of human feedback. This failure to explicitly model human quality further perpetuates the "teacher-quality ceiling" and hinders the agent's ability to surpass human performance.

To resolve this dilemma and enable agents to surpass imperfect human performance, we introduce VIQS that synergistically combines an advanced intervention mechanism with a sophisticated, quality-aware learning paradigm:

- **When to intervene**: We adopt a **value-based intervention** mechanism that proactively identifies when the agent's estimated value for its action significantly diverges from a more optimal (human) action. This focuses on strategic inferiority, granting the agent the essential freedom to explore and potentially discover superior actions, thereby enabling it to surpass its teacher.
- **How to learn**: We develop a novel **quality-aware value shaping** mechanism within a reward-free setting. Instead of blind trust, our approach leverages a GAN-inspired discriminator to dynamically assess the quality of human interventions and adaptively modulate the learning signal for the agent's Q-function. This explicitly models human action quality, allowing the agent to learn robustly from imperfect and low-frequency human guidance via temporal-difference (TD) propagation, requiring only a small number of interventions.

Our unified approach empowers the agent to explore freely, retain knowledge efficiently, and robustly handle imperfect human guidance, ultimately allowing it to outperform suboptimal experts.

**The main contributions of this paper are:**

1. VIQS, a new HIL-RL framework that effectively learns from sparse, imperfect human interventions in a reward-free manner, enabling the agent to outperform the guiding expert.
2. A novel, discriminator-guided quality-aware learning mechanism that dynamically assesses and adaptively incorporates expert feedback into the critic's value shaping.
3. Rigorous theoretical analysis highlighting our method's stability and its potential to surpass expert performance. In addition, extensive experiments in the MetaDrive simulator, demonstrating VIQS's superior performance compared to existing HIL-RL baselines, achieving higher safety and efficiency with fewer human interventions.

## 2 RELATED WORK

Our work builds upon and distinguishes itself from several key lines of research in learning from human feedback.

**Human-in-the-Loop Reinforcement Learning (HIL-RL).** HIL-RL is a distinct paradigm that focuses on integrating real-time, online human guidance directly into an agent's training process (Celemin et al., 2022). Unlike offline preference-based methods like RLHF (Christiano et al., 2017), which learn from static datasets of human judgments, HIL-RL emphasizes active, sequential inter-

vention where a human expert can directly take control or provide demonstrations during the agent's exploration (Celemin et al., 2022; Saunders et al., 2018). This paradigm is particularly crucial for safety-critical applications (e.g., autonomous driving, robotics) (Kelly et al., 2019; Peng et al., 2021), as it allows for immediate correction of dangerous behaviors, dramatically reducing the risks of unsafe exploration and obviating the need for meticulous reward engineering. For instance, recent systems like HIL-SERL (Luo et al., 2024) have demonstrated remarkable sample efficiency in complex, real-world robotic manipulation. However, its framework is predicated on a near-optimal expert, aiming to refine policy rather than surpass a sub-optimal teacher—the core challenge our work addresses. Moreover, our reward-free design circumvents its reliance on an auxiliary reward classifier. Our work is situated within this domain of online, interactive HIL-RL, addressing its core challenge of learning effectively from sparse and potentially imperfect intervention signals.

**Intervention Mechanisms in HIL-RL.** The design of intervention triggers critically impacts HIL-RL efficiency. Existing mechanisms fall into three categories: **action-based** triggers that rely on behavioral heuristics (Kelly et al., 2019; Li et al., 2022b), **uncertainty-based** approaches that estimate epistemic uncertainty (Hoque et al., 2021), and more advanced **value-based** methods that assess strategic suboptimality (Xue et al., 2023; McMahan et al., 2024; Gokmen et al., 2023). Our method builds upon value-based approaches but introduces a key innovation: using an expert-derived Q-function as a reference to estimate potential strategic inferiority, while explicitly accounting for its inherent variability through our quality-aware shaping mechanism.

**Learning from Imperfect and Sparse Guidance.** Recognizing that human guidance is rarely perfect is crucial (Park et al., 2024; Xue et al., 2025). GAIL (Ho & Ermon, 2016) draws an analogy between imitation learning and generative adversarial networks (GANs), in which a policy (generator) tries to mimic the expert's behavior, while a discriminator tries to differentiate between the expert's behavior and the policy's behavior. Our discriminator-based approach is inspired by this method, but we adapt it to assess quality rather than to mimic distributions. Most relevant to our work is the **PVP** family (Peng et al., 2023; 2025), which pioneered reward-free learning from interventions. However, PVP primarily relies on a trigger mechanism that reacts to human interventions, implying a direct correction to the agent's action rather than assessing strategic inferiority. Moreover, it assumes all interventions are equally optimal by using fixed proxy targets, overlooking the inherent imperfection and variability of human feedback. VIQS advances this line of work by integrating a value-based trigger and, most importantly, a quality-aware value shaping mechanism, enabling it to learn 'wisely' from imperfect experts.

## 3 PRELIMINARIES

Before detailing VIQS, we first define the problem setting and introduce the necessary background on actor-critic algorithms. We model the agent-environment interaction as a Markov Decision Process (MDP), defined by a tuple $\mathcal{M} = \langle \mathcal{S}, \mathcal{A}, P, \gamma, d_0 \rangle$, where $\mathcal{S}$ is the state space, $\mathcal{A}$ is the action space, $P(s'|s,a)$ is the transition function, $\gamma \in [0,1)$ is the discount factor, and $d_0$ is the initial state distribution. The agent's goal is to learn a policy $\pi$ that maximizes the expected discounted sum of rewards $\mathbb{E}_\pi[\sum_{t=0}^{\infty} \gamma^t r(s_t, a_t)]$. However, in our setting, the reward function $r(s,a)$ is **unknown** to the agent during training.

Our method builds upon the Twin Delayed Deep Deterministic Policy Gradient (TD3) algorithm (Fujimoto et al., 2018), a highly effective off-policy actor-critic algorithm for continuous control tasks. TD3 is renowned for its robust approach to mitigating overestimation bias in Q-value estimation, extending prior work like DDPG (Lillicrap et al., 2015) by employing twin critics and delayed policy updates. TD3 maintains an actor $\pi_\theta(s)$ and a pair of critic networks $\{Q_{\phi_1}, Q_{\phi_2}\}$, relying on a reward signal provided by the environment to guide its learning. The critics are trained to minimize the standard Bellman error based on environmental rewards, and the actor is updated by maximizing the Q-value from the more conservative critic. Our work fundamentally adapts this framework by modifying the critic's update rule to explicitly incorporate quality-aware expert guidance instead of environmental rewards, while the actor's update mechanism remains largely consistent.

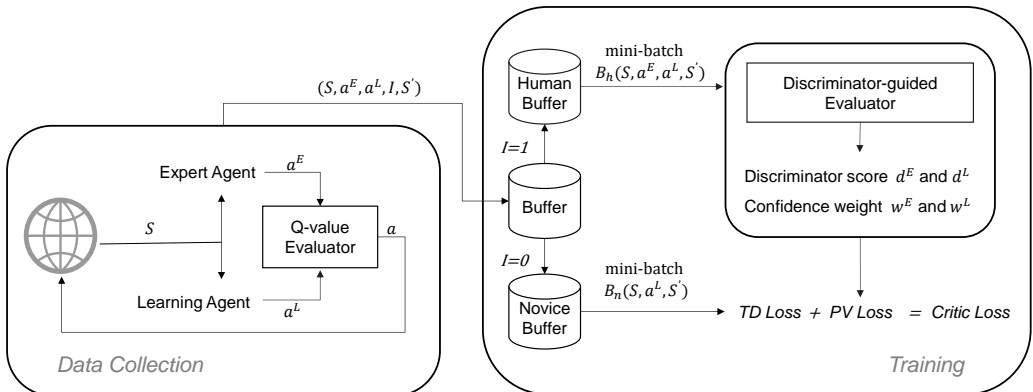

Figure 1: Pipeline of VIQS. **Left (Data Collection)**: A Q-value Evaluator dynamically selects between an expert's action ($a^E$) and the learning agent's action ($a^L$). Each transition $(S, a^E, a^L, I, S')$, including an intervention indicator $I$, is stored and then segregated into a **Human Buffer** ($B_h$) or **Novice Buffer** ($B_n$) respectively. **Right (Training)**: Data from $B_h$ trains a Discriminator-guided Evaluator, which provides quality scores ($d^E, d^L$) and confidence weights ($w^E, w^L$). The critic is then updated using a combined reward-free Temporal Difference (TD) loss and our novel Quality-aware Proxy Value (PV) loss, which leverages these quality signals from both $B_h$ and $B_n$ to learn effectively from imperfect human feedback and overcome the "teacher quality ceiling."

## 4 THE VIQS FRAMEWORK

VIQS overcomes the teacher-quality ceiling through two components (Figure 1): **value-guided intervention** for strategic intervention timing, and **quality-aware shaping** for robust learning from imperfect guidance. Unlike action-level imitation approaches, VIQS focuses on value optimization. The agent learns not what to do, but why certain actions are valuable, enabling it to discover superior strategies beyond expert demonstration while maintaining essential exploration autonomy.

### 4.1 WHEN TO INTERVENE: PROACTIVE, VALUE-BASED INTERVENTION

To break the teacher-quality ceiling, the agent must have the freedom to explore actions that are stylistically different from the expert's but strategically sound. We thus employ a **value-based trigger** instead of a naive action-difference trigger. This principle is illustrated in Figure 2. At each timestep $t$, we leverage a reference expert action-value estimator $Q^E$ (e.g., the critic of a pretrained expert policy $\pi^E$) to evaluate both the expert's proposed action $a_t^E$ and the learning agent's action $a_t^L$. An intervention is triggered only if the expert's action is deemed significantly more valuable:

$$I_t = \mathbb{I}\left(Q^E(s_t, a_t^E) - Q^E(s_t, a_t^L) > \tau_Q\right). \tag{1}$$

Here, $I_t$ is the intervention indicator, $\mathbb{I}(\cdot)$ is the indicator function, and $\tau_Q$ is a threshold. While $\tau_Q$ is a key hyperparameter governing the trade-off between agent autonomy and expert correction, our analysis in Appendix D.2 demonstrates that VIQS is robust to its choice. More importantly, we provide a principled heuristic for setting $\tau_Q$ based on the initial value gap, mitigating the need for extensive hyperparameter tuning. The terms $Q^E(s_t, a_t^E)$ and $Q^E(s_t, a_t^L)$ are the estimated Q-values for the expert's and agent's actions, respectively. For brevity in our discussion and figures, we denote this strategic value difference as $\Delta_Q = Q^E(s_t, a_t^E) - Q^E(s_t, a_t^L)$.

If $I_t = 1$, the expert's action $a_t^E$ is executed; otherwise, the agent's action $a_t^L$ is executed. In either case, the complete transition tuple $(s_t, a_t^E, a_t^L, s_{t+1}, I_t)$ is stored in the replay buffer.

### 4.2 HOW TO LEARN: VALUE SHAPING VIA QUALITY-AWARENESS

Having established **when** to intervene, we now detail **how** the agent learns from these critical interactions. We operate in a fully reward-free paradigm, deriving all learning signals directly from the intervention data itself.

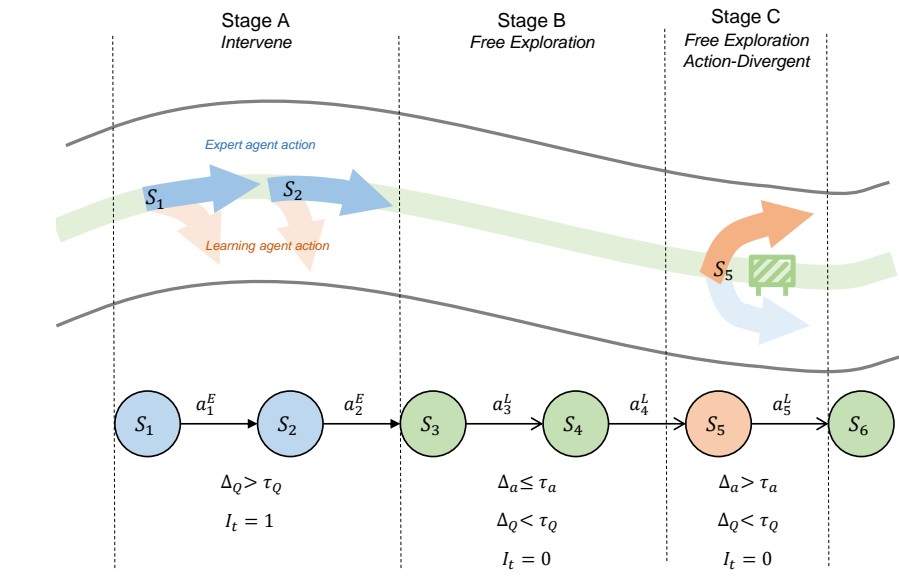

Figure 2: Illustration of VIQS's value-based intervention logic, which enables strategic exploration over simple mimicry. The diagram contrasts three critical scenarios: **(A) Necessary Intervention**: The agent proposes a strategically poor action. Since the value gap is large ($\Delta_Q > \tau_Q$), an intervention is triggered ($I_t = 1$) to prevent a mistake. **(B) Aligned Exploration**: The agent's action is strategically sound and aligns with the expert's, so it acts freely ($I_t = 0$). **(C) Superior Exploration**: This highlights the key advantage over action-based triggers. Although the agent's action diverges from the expert's, which a likelihood-based trigger would flag ($\Delta_a > \tau_a$), our system correctly identifies the action as strategically valid ($\Delta_Q \leq \tau_Q$) and withholds intervention. Here, $\Delta_a$ represents a measure of action divergence (e.g., negative log-likelihood of the agent's action under the expert policy $\pi^E$), and $\tau_a$ is a predefined threshold for such action-based intervention metrics.

### 4.2.1 DISCRIMINATOR-GUIDED QUALITY EVALUATION

To effectively learn from an imperfect expert, the agent must distinguish between high- and low-quality advice. We introduce a binary discriminator, $D_\psi(s, a)$, to serve this purpose. The discriminator is trained concurrently with the agent's policy from scratch, learning to distinguish the expert's actions from the agent's on intervention data ($I_t = 1$):

$$\mathcal{L}_{\text{disc}}(\psi) = -\mathbb{E}_{(s,a^E,a^L)\sim\mathcal{D}_{I=1}} \left[ \log D_\psi(s, a^E) + \log(1 - D_\psi(s, a^L)) \right]. \quad (2)$$

After a fixed number of training steps, the discriminator's parameters $\psi$ are **frozen**. It then functions as a stationary quality assessor for the remainder of the agent's training. Its output, $D_\psi(s, a)$, provides a consistent proxy for "expert-likeness." We transform the discriminator scores $d_t^E = D_\psi(s_t, a_t^E)$ and $d_t^L = D_\psi(s_t, a_t^L)$ into adaptive weights and dynamic proxy targets:

$$
\begin{aligned}
w_t^E &= d_t^E, & y_t^E &= 2d_t^E - 1, \\
w_t^L &= 1 - d_t^L, & y_t^L &= 2d_t^L - 1.
\end{aligned}
\quad (3)
$$

The weights $w_t$ serve as confidence scores, while the proxy targets $y_t \in [-1, 1]$ offer a much more nuanced shaping signal than the fixed targets used in prior work (Peng et al., 2023; 2025).

### 4.2.2 INTEGRATED CRITIC LOSS

The critic $Q_\phi$ is trained to minimize an integrated loss function that combines a standard reward-free TD-learning objective with our quality-aware proxy value (PV) term. The final critic loss is:

$$\mathcal{L}_{\text{critic}}(\phi) = \mathcal{L}_{\text{TD}}(\phi) + \alpha\mathcal{L}_{\text{PV}}(\phi), \quad (4)$$

where $\alpha$ is a balancing hyperparameter.

The PV loss acts only during interventions ($I_t = 1$), using the dynamically derived weights and targets from Equation 3 to shape the Q-function:

$$\mathcal{L}_{\text{PV}}(\phi) = \mathbb{E}_{(s,a^E,a^L) \sim \mathcal{D}_{I=1}} \left[ w_t^E (Q_\phi(s_t, a_t^E) - y_t^E)^2 + w_t^L (Q_\phi(s_t, a_t^L) - y_t^L)^2 \right]. \tag{5}$$

The TD loss ensures temporal consistency across all data:

$$\mathcal{L}_{\text{TD}}(\phi) = \mathbb{E}_{(s,a,s') \sim \mathcal{D}} \left[ Q_\phi(s,a) - \gamma \min_{i=1,2} Q_{\bar{\phi}_i}(s', \pi'(s')) \right]^2. \tag{6}$$

Here, following the TD3 framework, the target action $\pi'(s')$ is computed by adding clipped noise to the target policy's action: $\pi'(s') = \text{clip}(\bar{\pi}(s') + \epsilon, a_{low}, a_{high})$, where the noise is clipped $\epsilon \sim \text{clip}(\mathcal{N}(0, \tilde{\sigma}), -c, c)$. This target policy smoothing is a standard technique to prevent Q-value overestimation.

All expert guidance is integrated into the critic this way. The actor's update remains identical to the original TD3 formulation, learning implicitly by maximizing the well-shaped Q-value.

## 5 THEORETICAL ANALYSIS

In this section, we analyze the core mechanisms of VIQS, establishing the rationale for our design and proving its properties of stability and performance.

**Rationale for a Reward-Free Design.** Our reward-free framework directly addresses the *signal conflict* and *knowledge retention failure* common in methods that mix external rewards with sparse expert guidance. By making expert feedback the sole extrinsic signal, we ensure that knowledge injected via our PV loss is not overwritten by dense rewards. Instead, this knowledge is preserved and propagated by standard TD updates, guaranteeing a lasting impact.

**Grounding the Quality Signal.** Our use of a discriminator for quality assessment is formally grounded. At optimality, the discriminator learns the policy density ratio (Goodfellow et al., 2014): $D_\psi^*(s,a) = p_E(a|s)/(p_E(a|s) + \pi^L(a|s))$. Substituting this into our proxy target $y_t^E = 2d_t^E - 1$ reveals its true meaning:

$$y_t^E = \frac{p_E(a_t^E|s_t) - \pi^L(a_t^E|s_t)}{p_E(a_t^E|s_t) + \pi^L(a_t^E|s_t)}. \tag{7}$$

This shows $y_t^E$ is a **normalized policy density contrast**. This single, data-driven metric allows us to implicitly filter expert advice: it is strongly positive ($y_t^E \to 1$) for superior advice ($p_E \gg \pi^L$), negative for poor advice ($p_E < \pi^L$), and near-zero for redundant advice ($p_E \approx \pi^L$), all judged against the agent's current competence.

**Stability of Reward-Free Q-Learning.** Standard reward-free TD-learning can lead to Q-value divergence due to the lack of a grounding reward signal (Kumar et al., 2019). Our PV loss, $\mathcal{L}_{\text{PV}}$, prevents this by functioning as a powerful regularizer, analogous in spirit to Conservative Q-Learning (CQL) (Kumar et al., 2020). Expanding $\mathcal{L}_{\text{PV}}$ reveals its dual stabilizing mechanism:

$$\mathcal{L}_{\text{PV}}^{(s,a^x)} = w_t^x (Q_\phi(s_t, a_t^x) - y_t^x)^2 = \underbrace{-2 w_t^x Q_\phi(s_t, a_t^x) y_t^x}_{\text{(a) Value Anchoring}} + \underbrace{w_t^x (Q_\phi(s_t, a_t^x))^2}_{\text{(b) L2 Regularization}} + C. \tag{8}$$

This decomposition shows that $\mathcal{L}_{\text{PV}}$ simultaneously pulls the Q-function towards the bounded proxy targets $y_t^x \in [-1, 1]$ (Value Anchoring) and penalizes its magnitude via an explicit L2 term (L2 Regularization). Together, these two effects prevent both Q-value overestimation and unbounded growth, ensuring stable convergence. ehaved throughout training.

**Surpassing the Expert Teacher.** The ability to surpass the expert is enabled by our Actor-Critic design, where the actor maximizes value, not imitation. The critic learns a rich value landscape through a two-stage process. First, our $\mathcal{L}_{\text{PV}}$ acts as a **value anchor**, seeding the critic with foundational knowledge from the expert. Then, the standard $\mathcal{L}_{\text{TD}}$ loss acts as a **value propagator**. More

precisely, this propagation occurs by backing up values temporally along experienced trajectories (e.g., from state $s_t$ to $s_{t-1}$). Crucially, it is the inherent generalization capability of the neural network critic that extends this trajectory-specific knowledge to nearby, similar states. The synergy between temporal propagation via $\mathcal{L}_{\text{TD}}$ and spatial generalization via the function approximator is what allows the value function to be estimated over a wider portion of the state space. The stability and successful convergence of this entire value learning process are empirically demonstrated in Appendix E. With this learned Q-function, the actor then simply performs policy improvement. When our permissive trigger allows a novel, superior action that leads to a high-value state on this propagated map, the actor's optimization naturally discovers and selects it, enabling the agent's performance to exceed its teacher's.

## 6 EXPERIMENTS

Our empirical evaluation is designed to rigorously validate VIQS's central claim: overcoming the teacher-quality ceiling. We structure our experiments around four key research questions (RQs) that directly address the challenges outlined in Section 1:

1. **RQ1 (Performance & Ceiling Breakthrough):** Does VIQS consistently outperform state-of-the-art HIL methods and, most critically, break the teacher-quality ceiling across a spectrum of expert proficiencies?
2. **RQ2 (Data Efficiency):** How efficiently does VIQS learn compared to standard RL algorithms that require orders-of-magnitude more environmental interaction?
3. **RQ3 (Ablative Analysis):** What are the individual contributions of our core design principles: value-guided intervention and quality-aware shaping? Is the reward-free paradigm essential?
4. **RQ4 (Generalizability):** Can our quality-aware shaping mechanism serve as a general, plug-and-play module to enhance existing HIL frameworks?

### 6.1 EXPERIMENTAL SETUP

**Environment and Baselines.** We conduct all experiments in the challenging METADRIVE autonomous driving simulator (Li et al., 2022a). We benchmark VIQS against leading HIL methods (**HACO** (Li et al., 2022b), **TS2C** (Xue et al., 2023), **PVP** (Peng et al., 2023), **PVP4REAL** (Peng et al., 2025), and **HIL-SERL** (Luo et al., 2024)) and established from-scratch RL algorithms (**SAC**, **PPO**, and **TD3**). Further details on the environment configuration and task definitions are provided in Appendix A.1.

**Training Protocol.** To simulate a practical, data-constrained scenario, all HIL algorithms (including all variants for our ablation and generalizability studies) are trained for a concise 40K environment steps. In contrast, standard RL baselines are trained for a full 1M steps to establish an asymptotic performance benchmark. For statistical robustness, all reported results are the mean and standard deviation over 3 independent random seeds. For each run, we select the best performing model based on validation metrics and report its final performance on a completely held-out test set of unseen traffic scenarios.

**Implementation Details.** For a fair comparison within our shared actor-critic framework, we adapt TS2C to use a single Q-network as per the original authors' guidance for such cases (detailed in Appendix A.3). Furthermore, we include HIL-SERL, a state-of-the-art method from robotic manipulation, as a strong cross-domain baseline. To evaluate its effectiveness under our framework's core challenge—learning from sub-optimal human guidance—we made specific adaptations regarding its demonstration data and intervention strategy. A detailed description of this configuration, along with an in-depth analysis explaining its performance, is provided in Appendix G.

Complete details on network architectures and hyperparameters for all methods are cataloged in Appendix A.5. The corresponding learning curves for all experiments can be found in Appendix C.

**Simulated Experts of Graded Proficiency.** To systematically investigate the teacher-quality ceiling (RQ1), we employ pre-trained SAC agents to simulate experts at three distinct, graded proficiency levels: **High** (expert return $\approx 350$), **Medium** (return $\approx 270$), and **Low** (return $\approx 190$). The specifics of this controlled setup are detailed in Appendix A.2.

## 6.2 COMPARISON WITH HIL BASELINES (RQ1)

Table 1: Main comparison results. VIQS not only achieves state-of-the-art results but also consistently surpasses the performance of its guiding expert across all quality tiers, demonstrating its ability to break the teacher-quality ceiling.

| Expert Quality | Method | Training | Testing | | |
|---|---|---|---|---|---|
| | | Human Data Usage ($\downarrow$) | Episodic Return ($\uparrow$) | Episodic Safety Cost ($\downarrow$) | Success Rate (%) ($\uparrow$) |
| *High Expert (Return $\approx$ 350)* | | | | | |
| | SAC / Expert Policy | - | 348.9 | 0.76 | 74.0 |
| | HIL-SERL | 28.1K $\pm$ 4.4K (0.70) | 72.5 $\pm$ 100.6 | 18.5 $\pm$ 30.3 | 0.0 $\pm$ 0.0 |
| | HACO | 20.1K $\pm$ 0.4K (0.50) | 127.9 $\pm$ 23.1 | 37.5 $\pm$ 62.4 | 4.0 $\pm$ 6.9 |
| | TS2C | 15.0K $\pm$ 0.7K (0.38) | 90.8 $\pm$ 9.0 | 2.8 $\pm$ 1.3 | 0.0 $\pm$ 0.0 |
| | PVP | 21.3K $\pm$ 0.1K (0.53) | 294.3 $\pm$ 6.5 | **1.2 $\pm$ 0.7** | 54.0 $\pm$ 5.3 |
| | PVP4REAL | 18.3K $\pm$ 0.1K (0.46) | 348.2 $\pm$ 13.2 | 1.8 $\pm$ 0.9 | 75.3 $\pm$ 6.4 |
| | **VIQS (Ours)** | **7.6K $\pm$ 1.4K (0.19)** | **357.3 $\pm$ 20.7** | 2.1 $\pm$ 1.7 | **86.7 $\pm$ 6.4** |
| *Medium Expert (Return $\approx$ 270)* | | | | | |
| | SAC / Expert Policy | - | 266.7 | 0.7 | 36.0 |
| | HIL-SERL | 30.7K $\pm$ 1.6K (0.77) | 75.5 $\pm$ 26.9 | 3.0 $\pm$ 3.5 | 0.0 $\pm$ 0.0 |
| | HACO | 18.4K $\pm$ 1.2K (0.46) | 106.6 $\pm$ 23.7 | 2.9 $\pm$ 1.2 | 0.0 $\pm$ 0.0 |
| | TS2C | 14.7K $\pm$ 1.6K (0.37) | 99.9 $\pm$ 16.0 | 5.9 $\pm$ 4.9 | 0.7 $\pm$ 1.2 |
| | PVP | 19.6K $\pm$ 0.4K (0.49) | 242.1 $\pm$ 34.7 | 31.7 $\pm$ 52.7 | 30.7 $\pm$ 11.0 |
| | PVP4REAL | 15.9K $\pm$ 1.0K (0.40) | 296.4 $\pm$ 45.3 | 1.3 $\pm$ 0.4 | 54.7 $\pm$ 11.7 |
| | **VIQS (Ours)** | **4.1K $\pm$ 0.3K (0.10)** | **334.7 $\pm$ 13.5** | **0.6 $\pm$ 0.5** | **70.7 $\pm$ 10.3** |
| *Low Expert (Return $\approx$ 190)* | | | | | |
| | SAC / Expert Policy | - | 186.3 | 0.76 | 16.0 |
| | HIL-SERL | 31.4K $\pm$ 0.4K (0.79) | 63.5 $\pm$ 16.5 | 7.8 $\pm$ 11.6 | 0.0 $\pm$ 0.0 |
| | HACO | 22.8K $\pm$ 1.1K (0.57) | 99.3 $\pm$ 19.6 | 18.3 $\pm$ 29.4 | 2.7 $\pm$ 4.6 |
| | TS2C | 17.1K $\pm$ 1.1K (0.43) | 115.9 $\pm$ 26.0 | 9.7 $\pm$ 7.6 | 3.3 $\pm$ 5.8 |
| | PVP | 21.9K $\pm$ 0.1K (0.55) | 212.4 $\pm$ 14.5 | 84.9 $\pm$ 73.2 | 27.3 $\pm$ 3.1 |
| | PVP4REAL | 15.0K $\pm$ 1.4K (0.38) | **270.5 $\pm$ 15.4** | 1.0 $\pm$ 0.4 | **46.7 $\pm$ 3.1** |
| | **VIQS (Ours)** | **4.0K $\pm$ 0.9K (0.10)** | 265.4 $\pm$ 5.1 | **0.6 $\pm$ 0.1** | 38.7 $\pm$ 4.6 |

Answering RQ1, Table 1 shows that VIQS consistently surpasses its guiding expert, breaking the quality ceiling that constrains prior methods. The performance of the baselines reveals distinct limitations. For instance, both HACO and TS2C struggle significantly under sub-optimal guidance, resulting in near-zero success rates across all tiers. HIL-SERL, despite its efficacy in its native domain, also fails to learn effectively in our setup. As analyzed in Appendix G, this is due to a fundamental mismatch, as its learning paradigm is highly dependent on near-optimal expert guidance, which is not available here.

The limitations of these approaches become most apparent when facing a high-quality expert. Here, PVP4REAL's performance (348.2) is strictly capped by the expert's proficiency (348.9), clearly hitting the teacher-quality ceiling. In stark contrast, **VIQS is the only method to decisively and consistently break this barrier across all expert tiers**, achieving a new state-of-the-art return (357.3) with the high-quality expert. This demonstrates a crucial shift from incidental outperformance to a systematic discovery of novel, superior solutions.

While other strong baselines like PVP and PVP4REAL are less conservative and demonstrate the potential to exceed their guiding experts in certain scenarios—for example, both surpass the low-quality expert, and PVP4REAL surpasses the medium-quality one—their ability to do so appears inconsistent. PVP, for instance, fails to outperform the medium expert. This suggests that while promising, their learning mechanism does not guarantee a systematic breakthrough.

This superiority extends to a smarter trade-off between performance, safety, and efficiency across all expert levels. For instance, with the *Low* expert, VIQS achieves a competitive return (265.4) but with **4x less human data** and better safety (0.6 vs. 1.0 cost) than PVP4REAL. Conversely, while PVP can achieve a low safety cost (1.2) with a *High* expert, its performance collapses; VIQS maintains top-tier returns without excessive risk. Critically, across all settings, VIQS achieves its results with a **2.4x-4x reduction in human interventions**, highlighting an unparalleled return on guidance.

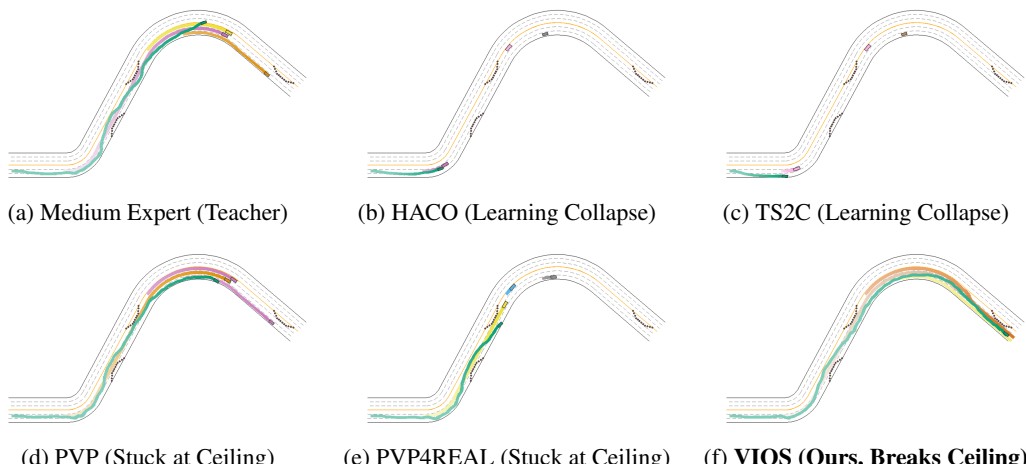

(a) Medium Expert (Teacher)  (b) HACO (Learning Collapse)  (c) TS2C (Learning Collapse)

(d) PVP (Stuck at Ceiling)  (e) PVP4REAL (Stuck at Ceiling)  (f) **VIQS (Ours, Breaks Ceiling)**

Figure 3: **Visualizing the Ceiling Breakthrough: A Qualitative Answer to RQ1.** Trajectories of HIL agents trained with the same suboptimal Medium expert (a). Baselines either fail to learn (b, c) or are trapped mimicking the teacher's flawed, high-variance path (d, e). In stark contrast, VIQS (f) learns a visibly smoother, more optimal policy, providing direct visual evidence of its ability to filter suboptimal advice and surpass its teacher.

This quantitative leap translates to qualitatively superior policies, as visualized in Figure 3. While baselines either forget prior knowledge or are trapped mimicking the teacher's suboptimal path, VIQS discovers a smoother and more direct trajectory. This visually confirms that our quality-aware mechanism successfully discerns and discards flawed guidance, liberating the agent to find a fundamentally better solution.

**Generalization to Classic Control Environments.** To verify that the advantages of VIQS extend beyond complex driving scenarios, we evaluate it on the classic `LunarLanderContinuous-v2` control benchmark. The results, detailed in Appendix F, reinforce our core findings. Despite being guided by a suboptimal expert (return $\approx$130), VIQS not only achieves a significantly higher final return (150.4) that surpasses its teacher, but does so with unparalleled efficiency. It requires less than half the expert interventions of PVP4REAL and one-third that of PVP, demonstrating the broad applicability of our value-guided intervention and quality-aware shaping principles.

## 6.3 A PARADIGM SHIFT IN SAMPLE EFFICIENCY (RQ2)

To answer RQ2, VIQS redefines sample efficiency when benchmarked against standard RL agents (Table 2). Guided by a high-quality expert, it matches the performance of the top baseline (TD3) using a mere 4% of the data (40k vs. 1M steps) while achieving a higher success rate (86.7% vs. 78.7%). This efficiency holds even with suboptimal guidance, where VIQS still outperforms strong agents like SAC and PPO with dramatically improved safety—up to a 10x lower cost than PPO. Overall, this $> 25\times$ efficiency gain establishes VIQS as a practical alternative to data-hungry standard RL.

Table 2: Sample efficiency comparison. VIQS achieves competitive performance with 4% of standard RL's data requirements.

| Method | Return ($\uparrow$) | Cost ($\downarrow$) | Succ. (% $\uparrow$) |
|---|---|---|---|
| *Standard RL Baselines (1M Steps)* | | | |
| SAC | 215.4±69.3 | 0.8±0.3 | 24.0±19.7 |
| PPO | 188.2±17.5 | 6.3±1.1 | 10.0±4.0 |
| TD3 | 356.9±3.2 | 2.2±1.8 | 78.7±2.3 |
| ***VIQS** with HIL Guidance (40k Steps)* | | | |
| + High Expert | 357.3±20.7 | 2.1±1.7 | 86.7±6.4 |
| + Medium Expert | 334.7±13.5 | **0.6±0.5** | 70.7±10.3 |
| + Low Expert | 265.4±5.1 | **0.6±0.1** | 38.7±4.6 |

## 6.4 ABLATION AND GENERALIZABILITY ANALYSIS (RQ3 & RQ4)

To isolate the contributions of each component (RQ3) and test the generalizability of our principles (RQ4), we conduct a series of targeted experiments. Results are summarized in Tables 3 and 4.

Table 3: Ablation study of VIQS's components. We evaluate variants with specific mechanisms disabled or altered, using the medium-quality expert and 40k training steps.

| | Training | Testing | | |
|---|---|---|---|---|
| Variant | Human Data Usage (↓) | Return (↑) | Safety Cost (↓) | Success (%) (↑) |
| **VIQS (Full)** | **4.1K ± 0.3K (0.10)** | **334.7 ± 13.5** | **0.6 ± 0.5** | **70.7 ± 10.3** |
| *Ablated Variants* | | | | |
| w/o Shaping | 4.8K ± 1.5K (0.12) | 301.5 ± 30.5 | 10.0 ± 15.9 | 57.0 ± 13.9 |
| + Env. Reward | 8.5K ± 3.9K (0.21) | 206.1 ± 64.1 | 2.5 ± 1.3 | 20.3 ± 2.2 |
| Action-based Trigger | 27.3K ± 2.3K (0.68) | 86.6 ± 31.4 | 21.5 ± 27.1 | 1.3 ± 0.2 |

Table 4: Generalization of our framework. We apply our core principles to enhance several existing HIL methods. The '+' indicates a baseline algorithm augmented by our contributions.

| | Training | Testing | | |
|---|---|---|---|---|
| Method | Human Data Usage (↓) | Return (↑) | Safety Cost (↓) | Success (%) (↑) |
| HACO | 18.4K ± 1.2K (0.46) | 106.6 ± 23.7 | 2.9 ± 1.2 | 0.0 ± 0.0 |
| **HACO+** | **9.5K ± 0.8K (0.23)** | **172.0 ± 90.8** | **2.9 ± 1.6** | **17.3 ± 30.0** |
| PVP | 19.6K ± 0.4K (0.49) | 242.1 ± 34.7 | 31.7 ± 52.7 | 30.7 ± 11.0 |
| **PVP+ (VIQS)** | **4.1K ± 0.3K (0.10)** | **334.7 ± 13.5** | **0.6 ± 0.5** | **70.7 ± 10.3** |
| PVP4REAL | 15.9K ± 1.0K (0.40) | 296.4 ± 45.3 | 1.3 ± 0.4 | 54.7 ± 11.7 |
| **PVP4REAL+** | **2.3K ± 0.3K (0.06)** | **316.8 ± 34.4** | **0.6 ± 0.2** | **62.7 ± 21.4** |

**Ablation Study (RQ3).** Our ablations (Table 3) confirm that each component of VIQS is critical. Removing quality-aware value shaping (w/o Shaping) cripples policy safety, increasing safety cost from 0.6 to 10.0. This shows that blindly trusting expert values induces reckless behavior. Further, adding environmental rewards (+ Env. Reward) is actively harmful; it dilutes the sparse human guidance, causing the success rate to plummet from 70.7% to 20.3% while doubling human data usage. Most strikingly, replacing our value-based trigger with an action-based one leads to a catastrophic failure: despite using $6.7\times$ more human data, the policy's success rate is a mere 1.3%. This highlights that intervention must be based on value discrepancy, not action mismatch, to be effective.

**Generalizability Study (RQ4).** Our framework's principles generalize broadly (Table 4). Our method, VIQS, is simply the PVP baseline enhanced with our principles, slashing its human data requirement by 79% while achieving state-of-the-art success. This efficiency gain is not isolated; applying the same principles reduces data needs for HACO and PVP4REAL by 48% and 85% respectively, with corresponding performance gains. Notably, PVP4REAL+, equipped with our module, matches its base performance with just 15% of the original human data (2.3k vs. 15.9k). These results confirm our framework acts as a plug-and-play enhancement to radically boost HIL efficiency. Further details are in Appendix B.

# 7    CONCLUSION

We presented **VIQS**, a reward-free HIL framework that learns effectively from imperfect human guidance. Its core lies in a novel combination of a value-based intervention trigger and quality-aware value shaping, a design that enables VIQS to consistently surpass the "teacher-quality ceiling" with high data efficiency. Crucially, our principles generalize: when applied to other HIL methods, they act as a plug-and-play enhancement, yielding significant performance gains. VIQS thus offers a robust and efficient blueprint for learning from flawed human input.

**Limitations and Future Work.** Our primary limitation lies in the experimental use of a pre-existing expert value function, $Q^E$, to trigger interventions. This setup was instrumental for a controlled and reproducible analysis of how an agent learns from imperfect guidance. However, a key strength and core design principle of our framework is the **decoupling of this intervention trigger from the quality-aware learning mechanism**. This inherent modularity is precisely what makes VIQS adaptable to real-world scenarios where an explicit $Q^E$ is unavailable.

This modularity opens up several practical deployment paths. For instance, the trigger can be replaced with $Q^E$**-free signals**, such as direct human commands (e.g., via a button press) or agent-based metrics like high model uncertainty. Once an intervention is triggered by any such method, our discriminator-based mechanism proceeds to assess the relative quality of the expert's and agent's actions, enabling the core benefit of VIQS—critical learning from guidance. Alternatively, one could **learn a proxy value function** to serve as the trigger. Inspired by advances in preference-based learning, this proxy could be trained offline on human judgments (e.g., pairwise comparisons via RLHF) to capture the expert's latent intent. Future work will focus on empirically validating these alternative triggers, while also exploring richer feedback modalities, such as natural language, and developing strategies for scenarios with budgeted human availability.

## ETHICS STATEMENT

This research adheres to the ICLR Code of Ethics. All experiments were conducted in publicly available simulation environments. To ensure reproducibility and mitigate ethical concerns, human experts were simulated using pre-trained AI agents. This "agent-as-expert" approach entirely avoids the collection of data from human subjects, thus aligning with principles of responsible research. Our work aims to contribute positively by developing safer and more reliable AI systems.

## GENERATIVE AI USAGE STATEMENT

During the preparation of this manuscript, large language models (LLMs) were used solely to refine the clarity, conciseness, and grammatical correctness of the text. All scientific content, including ideas, design, results, and interpretation, was conceived and verified by the authors, who retain full responsibility for the paper's intellectual content.

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

## A EXPERIMENTAL AND IMPLEMENTATION DETAILS

This section provides a comprehensive account of our experimental setup, including environment specifications, expert policy generation, baseline modifications, and a full list of hyperparameters, to ensure the reproducibility of our work.

### A.1 ENVIRONMENT AND TASK SPECIFICATION

All experiments are conducted in the **MetaDrive Safety Benchmark** (Li et al., 2022a), a driving simulator featuring procedurally generated environments that enables robust evaluation of safety and generalization.

**Training and Test Environments.** We leverage MetaDrive's procedural generation to create distinct sets of environments for training and testing. The training set consists of 50 unique maps. For final evaluation, we use a separate, held-out test set of 50 unseen maps, and the performance is averaged over 50 test episodes. This strict separation and evaluation protocol ensures a fair and robust assessment of each policy's generalization capability. In each episode, both the environment layout and initial vehicle placements are randomized to promote robust learning.

**State and Action Space.** The agent receives a low-dimensional state vector from the simulator, including ego-vehicle state (speed, heading, etc.), Lidar-like sensor readings, and navigation information. We use standard Multi-Layer Perceptron (MLP) networks for all policies and value functions. The action space is a continuous 2D vector `[steering, acceleration/braking]`, with components normalized to $[-1, 1]$.

**Task Reward Formulation.** Crucially, the following task reward function is **not available** to our method (VIQS) or other reward-free HIL baselines during training. It serves two specific purposes: 1) pre-training the expert policies, and 2) as the ground-truth metric for evaluating task performance of all methods during the final test phase. The episodic reward is composed of four distinct components:

$$R = c_{\text{disp}}R_{\text{disp}} + c_{\text{speed}}R_{\text{speed}} + c_{\text{collision}}R_{\text{collision}} + R_{\text{term}}. \tag{9}$$

The individual components are defined as follows:

- **Displacement Reward** ($R_{\text{disp}}$): A dense reward for longitudinal distance traveled towards the destination. $c_{\text{disp}} = 1.0$.
- **Speed Reward** ($R_{\text{speed}}$): A dense reward proportional to the agent's speed ($v_t/v_{\text{max}}$). $c_{\text{speed}} = 0.1$.
- **Collision Penalty** ($R_{\text{collision}}$): A large penalty of -5 incurred upon any collision event, meaning $c_{\text{collision}}R_{\text{collision}} = -5$.
- **Terminal Reward** ($R_{\text{term}}$): A sparse reward/penalty at the end of an episode: +10 for reaching the destination, -5 for driving off-road.

**Safety Cost Formulation.** To explicitly measure safety, we define a separate cost function based on discrete violation events. A cost of $c = 1.0$ is incurred upon the occurrence of a safety violation:

$$c_t = \begin{cases} 1.0 & \text{if out-of-road, crash-vehicle, or crash-object event occurs,} \\ 0.0 & \text{otherwise.} \end{cases} \tag{10}$$

This cost signal serves a dual purpose in our experimental framework:

- **For Evaluation:** During testing, the cumulative episodic cost, $\sum_t c_t$, serves as the primary metric for safety. It is kept separate from the task reward (Eq. 9) to provide a distinct and clear measure of a policy's safety performance, where lower is better.
- **For Training:** The cost signal is primarily utilized by some baselines during their training phase. Specifically, each instance of a safety violation (a cost event) is treated as an immediate negative reward of -1, which is incorporated into the agent's total reward for that timestep.

### A.2 EXPERT POLICY DETAILS

To systematically investigate the "teacher-quality ceiling," we generated expert policies at three distinct proficiency tiers: Low, Medium, and High. This was accomplished through a multi-stage

training and selection process designed to produce a well-defined spectrum of expert capabilities. The foundational algorithm for all experts was Soft Actor-Critic (SAC), implemented using the Stable-Baselines3 (SB3) library (Raffin et al., 2021).For consistency and reproducibility, all training runs for generating these experts were conducted with a fixed random seed (seed=0).

**Generation of Low and Medium-Quality Experts.** The Low and Medium experts were derived from a single, unified training run. We trained an SAC agent for 1 million timesteps in the standard training environment, which consists of 50 procedurally generated maps. Throughout this process, model checkpoints were saved periodically. From this single training trajectory, we selected two distinct experts:

- **Low-Quality Expert:** We selected the checkpoint saved at approximately **40,000 training steps**. This early-stage model represents a novice policy that has learned basic driving skills but is still prone to frequent errors and suboptimal decisions.
- **Medium-Quality Expert:** We identified the **best-performing checkpoint** from the entire 1-million-step training run, based on its evaluation performance during training. This model represents a competent but ultimately suboptimal expert, whose capabilities are constrained by the diversity of the initial 50-map training environment.

**Generation of the High-Quality Expert.** Achieving top-tier performance that could truly test the limits of our framework required an extended, continual training strategy. We found that the initial 1M-step training presented a performance bottleneck. To surpass this, we proceeded as follows: starting from the best-performing model of the initial 1M-step run (i.e., our Medium-Quality expert), we continued its training under an enhanced regime.

- The training environment was **expanded to 100 unique maps** to provide greater scenario diversity.
- The agent was then trained for an **additional 2 million timesteps**, bringing its total training exposure to 3 million steps.

The **best-performing checkpoint** from this extended 3M-step training trajectory was then designated as our High-Quality Expert.

**Final Empirical Validation.** Crucially, the proficiency tier of each expert was not merely assumed from its training history. Each of the three selected checkpoints was subjected to a final, rigorous validation on a held-out test set, which consists of 50 unseen maps. This same test set was used for the final evaluation of all algorithms in our paper. We averaged their performance over 50 test episodes, with one episode being run on each of the 50 unique test maps, to obtain a stable and objective measure of their true, generalizable skill. This empirical validation confirmed our three distinct quality levels, which correspond to the expert performance baselines reported in our main experimental results (Table 1):

- **High Expert:** Achieved an average test return of $\approx 350$.
- **Medium Expert:** Achieved an average test return of $\approx 270$.
- **Low Expert:** Achieved an average test return of $\approx 190$.

This multi-stage process ensures that our study is grounded in a controlled and empirically verified hierarchy of expert qualities, providing a solid foundation for evaluating whether an agent can surpass its teacher.

### A.3 MODIFICATIONS TO BASELINES FOR CONTROLLED COMPARISON

To create a rigorously controlled experimental setting that isolates the core learning mechanism of each algorithm, we adapted the original TS2C baseline (Xue et al., 2023) in two key aspects.

- **Standardization of the Teacher Architecture.** The original TS2C employs a "teacher" model built from an **ensemble of Q-networks**. To ensure that any observed performance differences stem from the agent's learning algorithm itself, rather than from variations in the expert model's architecture or capacity, we replaced this ensemble teacher. In our

Table 5: Hyperparameters for the SAC training run to generate expert candidates.

| Hyperparameter | Value |
|---|---|
| *Algorithm & Network* | |
| Algorithm | Soft Actor-Critic (SAC) |
| Policy Network | MlpPolicy |
| Network Architecture | 2 hidden layers, 256 units each, ReLU activation |
| *Training* | |
| Optimizer | Adam |
| Learning Rate | 1e-4 |
| Batch Size | 1024 |
| Buffer Size | 1,000,000 |
| Learning Starts | 10,000 steps |
| Discount Factor ($\gamma$) | 0.99 |
| Target Update Rate ($\tau$) | 0.005 |
| Gradient Steps | 1 |
| Train Frequency | 1 step |
| Entropy Coefficient (ent_coef) | Automatic tuning ('auto') |
| Target Entropy | Automatic tuning ('auto') |
| *Environment* | |
| Num. Parallel Envs | 4 ('SubprocVecEnv') |
| Num. Training Scenarios | 50 |

setup, all methods, including the modified TS2C, receive guidance from a **single, pre-trained SAC-based Q-network**. This standardizes the source of expert knowledge across all comparisons. The learning agent itself, for all baselines, utilizes a standard twin-critic architecture to mitigate Q-value overestimation.

- **Alignment of the Intervention Trigger.** We replaced TS2C's original intervention trigger, which is based on the agent's self-perceived value function uncertainty ($V^{\pi_t}(s) - \mathbb{E}_{a \sim \pi_s}[Q^{\pi_t}(s, a)] > \epsilon$), with the action-centric trigger used across our entire study: $Q(s, a_h) - Q(s, a_n) > \tau_q$. This alignment ensures that all agents request and receive help under **identical conditions**, based on the same principle of detectable action superiority.

**Rationale for Modifications.** These modifications are essential for isolating the central scientific question of this paper: how effectively different algorithms **learn from** sparse human guidance. The original TS2C's trigger (based on *value function instability*) and our study's trigger (based on the *comparative superiority of an expert action*) represent fundamentally different intervention philosophies.

By standardizing both the teacher architecture and the intervention trigger, we create an experimental design that meticulously **controls for critical confounding variables**. These include (1) the representational capacity of the teacher model and (2) the philosophical basis for when and why interventions occur. Consequently, any resulting performance differences can be confidently attributed to the algorithms' intrinsic mechanisms for **processing and internalizing the guidance itself**. This principled approach enables a direct, fair, and interpretable comparison of the core learning strategies at the heart of our work.

## A.4 NETWORK ARCHITECTURES

For consistency and fair comparison, all algorithms, including our method (VIQS), use a standard Multi-Layer Perceptron (MLP) architecture for their respective policy and value networks.

Specifically, unless otherwise noted, all actor and critic networks consist of **two hidden layers with 256 units each**. Each hidden layer is followed by a **Rectified Linear Unit (ReLU)** activation function. This architecture is consistent across VIQS, HACO, PVP, VIQS, TS2C, SAC, TD3, and

PPO, as derived from their SB3 `MlpPolicy` implementations. It is worth noting that for HACO, we follow its original implementation where the actor and critic networks do not share features (`share_features_extractor=False`). For all other algorithms, we use the standard SB3 defaults for feature extraction.

## A.5 HYPERPARAMETERS

We divide the hyperparameters into three tables for clarity. Table 6 lists parameters shared across all off-policy HIL baselines. Table 7 details the crucial differences between these HIL methods. Finally, Table 8 outlines the hyperparameters for the standard RL agents trained from scratch.

Table 6: Hyperparameters shared across all HIL off-policy methods (VIQS, HACO, PVP, TS2C, etc.).

| Hyperparameter | Value |
|---|---|
| Discount Factor ($\gamma$) | 0.99 |
| Target Update Rate ($\tau$) | 0.005 |
| Replay Buffer Size | 50,000 |
| Optimizer | Adam |
| Train Frequency | 1 step |

Table 7: Specific hyperparameters for HIL methods, highlighting key differences in their intervention mechanism and learning strategy. Note that only PVP4REAL uses an explicit BC loss.

| Hyperparameter | VIQS (Ours) | HACO | PVP / VIQS | PVP4REAL | TS2C |
|---|---|---|---|---|---|
| *Intervention Mechanism* | | | | | |
| Action Intervention Threshold ($\alpha_{\text{free}}$) | - | 0.95 | 0.95 | 0.95 | - |
| Value Intervention Threshold ($q_{\text{free}}$) | 1.0 | - | - | - | 0.5 |
| *Algorithm & Learning* | | | | | |
| Base Policy | TD3-based | SAC-based | TD3-based | TD3-based | SAC-based |
| Learning Rate | 1e-4 | 1e-4 (Actor/Critic/Ent) | 1e-4 | 1e-4 | 1e-4 |
| Batch Size | 1024 | 1024 | 1024 | 1024 | 1024 |
| Learning Starts | 100 | 100 | 10 | 10 | 10 |
| BC Loss Weight ($\lambda_{\text{BC}}$) | **0** | 0 | 0 | **1.0** | 0 |
| Entropy Coef. (`ent_coef`) | N/A | 'auto' | N/A | N/A | 'auto' |

Table 8: Hyperparameters for standard RL baselines trained from scratch for 1M steps.

| Hyperparameter | SAC | TD3 | PPO |
|---|---|---|---|
| Learning Rate | 1e-4 | 1e-4 | 5e-5 |
| Batch Size | 1024 | 100 | 256 |
| Buffer Size | 1,000,000 | 1,000,000 | N/A |
| Learning Starts | 10,000 | 10,000 | N/A |
| *PPO-Specific Parameters* | | | |
| Num. Steps (`n_steps`) | N/A | N/A | 1024 |
| Num. Epochs (`n_epochs`) | N/A | N/A | 500 |
| Clip Range | N/A | N/A | 0.1 |
| Value Func. Coef. (`vf_coef`) | N/A | N/A | 0.5 |
| Entropy Coef. (`ent_coef`) | N/A | N/A | 0.0 |
| Max Grad Norm | N/A | N/A | 10.0 |

## B GENERALIZABILITY OF THE PROPOSED PRINCIPLE

We claim that quality-aware value shaping, our core principle, is a highly generalizable and modular component. To prove its versatility, we integrate VIQS's core *Value-Shaping Module* into

three distinct existing learning algorithms: **HACO** (Li et al., 2022b), **PVP** (Peng et al., 2023), and **PVP4REAL** (Peng et al., 2025). The goal is to demonstrate that our module can act as a universal upgrade, enhancing the performance, safety, and efficiency of various human-in-the-loop and imitation learning approaches.

**Integration Method Overview.** The core idea behind our integration is to augment or, in specific cases, replace the value estimation mechanism within each baseline algorithm with our Quality-Aware Value Shaping. Instead of directly relying on the baseline's inherent reward processing, our module intelligently guides what the agent perceives as "good" or "bad" actions, thereby steering its learning trajectory away from expert flaws. This makes the integration clean and minimally invasive, highlighting the module's plug-and-play nature. Below, we detail the integration into HACO, PVP, and PVP4REAL.

### B.1 INTEGRATION WITH HACO (HACO+)

The Human-AI Copilot Optimization (HACO) algorithm (Li et al., 2022b) employs a dual-value-function approach to incorporate human guidance. It learns a standard Q-function, $Q(s, a)$, for task returns, and a separate intervention value function, $Q^{\text{int}}(s, a)$, to estimate the long-term cost of deviating from expert suggestions. The final policy is then optimized to maximize $Q$ while minimizing $Q^{\text{int}}$.

Our integration, HACO+, introduces a fundamental redesign of HACO's learning dynamics. Instead of maintaining two separate and potentially conflicting objectives for the actor, we propose a streamlined framework that unifies guidance and task learning within a single, quality-aware Q-function. This is achieved through two key modifications based on our discriminator $D_\psi(s, a)$.

**Quality-Aware Critic Shaping.** We replace HACO's original auxiliary critic loss with a more powerful, discriminator-weighted term reminiscent of Conservative Q-Learning (CQL). At each intervention step ($I_t = 1$), we actively shape the Q-function's landscape by encouraging it to assign higher values to high-quality expert actions ($a_h$) and lower values to the concurrent, lower-quality agent actions ($a_n$). The total critic loss for each critic $Q_\phi$ in the ensemble is:

$$\mathcal{L}_{\text{critic}}^{\text{HACO+}}(\phi) = \mathcal{L}_{\text{TD}}(\phi) - \lambda_{\text{cql}} \cdot \mathbb{E}_{(s_t, a_h, a_n) \sim \mathcal{B}_I} \left[ w_h Q_\phi(s_t, a_h) - w_n Q_\phi(s_t, a_n) \right], \quad (11)$$

where $\mathcal{L}_{\text{TD}}(\phi)$ is the standard Mean Squared Bellman Error, and $\mathcal{B}_I$ is the buffer of intervention data. The weights $w_h$ and $w_n$ are derived from our discriminator:

- $w_h = D_\psi(s_t, a_h)$ represents the perceived quality of the expert's action.
- $w_n = 1 - D_\psi(s_t, a_n)$ represents the "non-quality" or inferiority of the agent's action rejected by the human.

By minimizing $\mathcal{L}_{\text{critic}}^{\text{HACO+}}$, the optimizer seeks to maximize the term $w_h Q_h - w_n Q_n$, effectively pushing up Q-values for good actions and suppressing them for bad ones, weighted by the discriminator's confidence. This embeds the expert's knowledge directly and dynamically into the primary value function.

**Decoupled Actor Objective.** A direct consequence of our expressive critic shaping is the simplification of the actor's objective. Since all guidance information is now encoded within the main Q-function, we can completely decouple the actor from the specialized cost-critic $Q^{\text{int}}$. The actor is no longer required to balance two objectives and is free to optimize a standard, entropy-regularized policy objective:

$$\max_\theta \mathbb{E}_{s_t \sim \mathcal{B}, a_t \sim \pi_\theta} [\min_i Q_{\phi_i}(s_t, a_t) - \alpha \log \pi_\theta(a_t | s_t)]. \quad (12)$$

Here, the actor simply trusts the (ensembled minimum of) shaped Q-functions learned via Eq. 11. By removing the explicit penalty term $-Q^{\text{int}}(s, a)$, we liberate the agent from the constraint of long-term imitation. This allows the policy to fully exploit the learned value landscape and discover novel strategies that may significantly outperform the human expert, a goal that is fundamentally restricted by HACO's original formulation. While we still train the $Q^{\text{int}}$ network as part of the overall critic loss for stable learning, its output is not used in the final actor update.

In essence, HACO+ transforms HACO from a constrained, dual-objective optimization problem into a more elegant, single-objective framework where expert guidance serves to shape a unified value function for more effective policy improvement.

## B.2 INTEGRATION WITH PVP (PVP+)

The original PVP (Proxy Value Policy Optimization) algorithm (Peng et al., 2023) is a reward-free HIL-RL method that assigns fixed proxy values to human interventions. Specifically, when an intervention occurs ($I_t = 1$), PVP sets a target value for the Q-function. Its primary goal is to learn from these interventions to align the agent's policy with expert demonstrations. Our VIQS framework, as presented in Section 4, is built upon the foundational idea of reward-free learning from interventions, just like PVP. However, VIQS critically enhances this by introducing:

- A **proactive, value-based intervention trigger** (Eq. 1) that focuses on strategic inferiority, allowing the agent to explore and surpass the expert.
- A **discriminator-guided quality-aware value shaping mechanism** (Eqs. 2, 3, 5) that dynamically assesses human intervention quality and modulates the Q-function's learning signal. This replaces PVP's fixed proxy targets with context-aware, quality-dependent values.

Therefore, integrating our 'Value-Shaping Module' into PVP effectively transforms it into our full VIQS method. The 'PVP+' in our experiments *is* precisely VIQS, demonstrating how our innovations overcome PVP's limitation of blindly trusting human feedback and employing a simpler intervention trigger.

## B.3 INTEGRATION WITH PVP4REAL (PVP4REAL+)

PVP4REAL (Peng et al., 2025) extends PVP by incorporating an additional behavioral cloning (BC) loss term into the actor's objective. This BC loss directly encourages the agent's policy to mimic the expert's actions, aiming to further stabilize training and improve efficiency, especially in complex real-world scenarios. When we integrate our 'Value-Shaping Module' into PVP4REAL, we adapt our critic-shaping mechanism into its reward-free framework, similar to how 'PVP+' becomes VIQS. However, for 'PVP4REAL+', we retain the original BC loss on the actor. The actor's objective for 'PVP4REAL+' therefore becomes:

$$\max_{\theta} \mathbb{E}_{s_t \sim \mathcal{B}, a_n \sim \pi_n(\cdot|s_t;\theta)}[\min_i Q_i(s_t, a_n) - \alpha \log \pi_n(a_n|s_t;\theta)] + \lambda_{\mathrm{BC}} \mathcal{L}_{\mathrm{BC}}(\theta), \qquad (13)$$

where $\mathcal{L}_{\mathrm{BC}}(\theta)$ is the behavioral cloning loss between the agent's policy $\pi_n$ and the expert's policy $\pi_E$ (or expert actions $a_E$), typically an MSE loss for continuous actions or a cross-entropy loss for discrete actions, and $\lambda_{\mathrm{BC}}$ is its weighting coefficient. The critic's updates for 'PVP4REAL+' are identical to those in VIQS, incorporating the discriminator-guided quality-aware value shaping. This integration demonstrates that our shaping module is compatible with methods that also utilize auxiliary imitation losses on the actor.

# C ADDITIONAL RESULTS AND LEARNING CURVES

This section provides the complete learning curves for all experiments discussed in the main paper. These plots complement the final performance tables by illustrating the learning dynamics, sample efficiency, and stability of the algorithms throughout the training process. All curves depict the mean performance over three random seeds, with the shaded regions representing the standard deviation. Performance is evaluated periodically on the held-out test set.

## C.1 COMPARISON WITH HIL BASELINES ACROSS EXPERT TIERS

Figures 4, 5, and 6 present the learning curves for VIQS and all HIL baselines when guided by the High-, Medium-, and Low-Quality experts, respectively. These results visually corroborate our core claim (RQ1). Across all expert tiers, VIQS demonstrates a significantly steeper and more stable learning trajectory, rapidly achieving high performance and consistently surpassing the expert's own performance level (the dashed line). This stands in stark contrast to other baselines, which either exhibit instability, learn much slower, or clearly plateau at or below the teacher-quality ceiling.

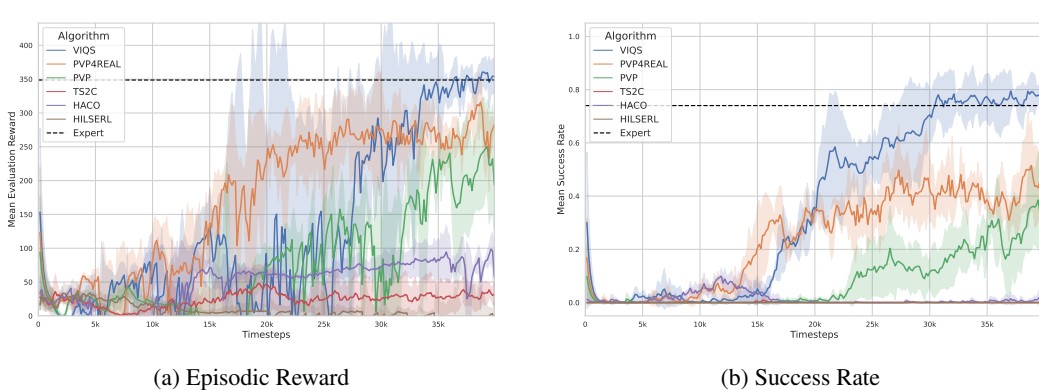

(a) Episodic Reward

(b) Success Rate

Figure 4: Learning curves for HIL baselines guided by the **High-Quality Expert** (Return ≈ 350). VIQS quickly surpasses the expert's performance.

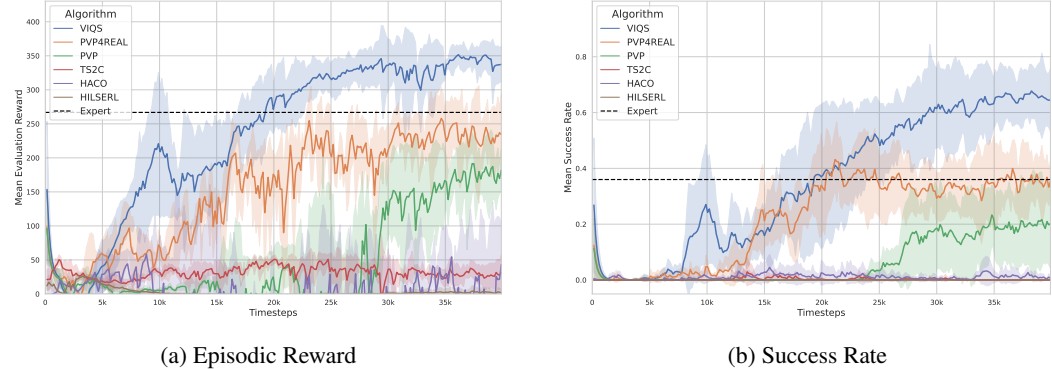

(a) Episodic Reward

(b) Success Rate

Figure 5: Learning curves for HIL baselines guided by the **Medium-Quality Expert** (Return ≈ 270). VIQS again shows superior learning efficiency and breaks the ceiling.

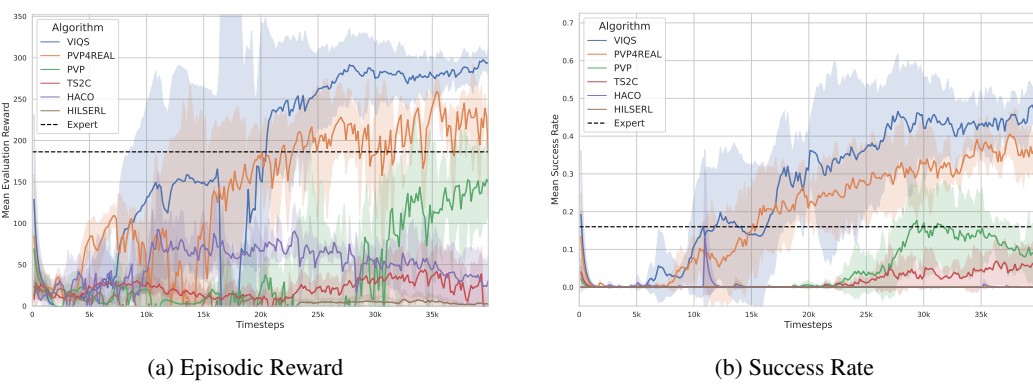

(a) Episodic Reward

(b) Success Rate

Figure 6: Learning curves for HIL baselines guided by the **Low-Quality Expert** (Return ≈ 190). Even with poor guidance, VIQS learns effectively and significantly outperforms its teacher.

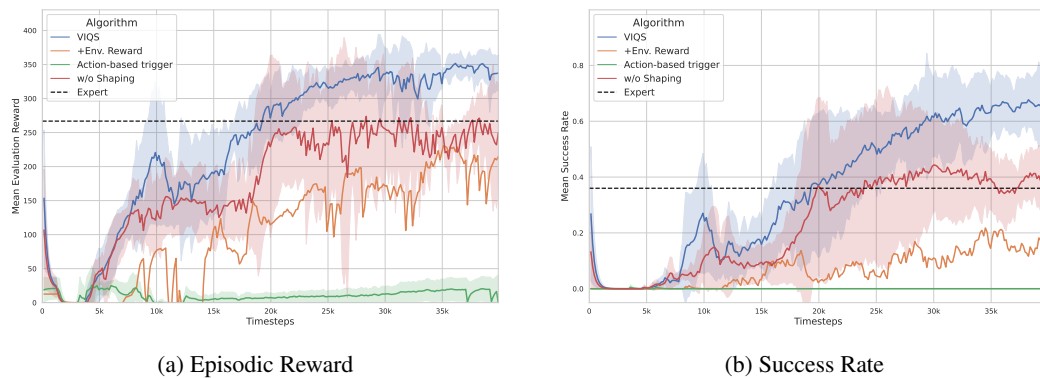

(a) Episodic Reward

(b) Success Rate

Figure 7: Learning curves for the ablation study of VIQS. The full model ('VIQS') significantly outperforms all ablated variants, confirming the importance of its core components.

## C.2 ABLATION STUDY OF VIQS COMPONENTS

Figure 7 provides the learning curves for our ablation study (RQ3), conducted with the Medium-Quality expert. These plots visually demonstrate the criticality of each component of VIQS. Removing quality-aware shaping ('w/o Shaping') leads to unstable and inferior performance. Re-introducing environmental rewards ('+Env. Reward') severely impedes learning, showing a flat curve near zero. Most dramatically, reverting to a naive 'Action-based trigger' results in a complete learning failure. These curves underscore that the synergy between our value-guided intervention and quality-aware shaping is essential for success.

## C.3 GENERALIZABILITY STUDY OF THE VIQS FRAMEWORK

Figures 8, 9, and 10 showcase the results of our generalizability study (RQ4). We applied our core quality-aware shaping mechanism to enhance three existing HIL algorithms: HACO, PVP, and PVP4REAL. In each case, the enhanced version ('+') demonstrates a marked improvement in learning speed, stability, and final performance compared to its original counterpart. This provides strong visual evidence that our proposed mechanisms act as a general, plug-and-play module to significantly boost the performance and efficiency of various HIL frameworks.

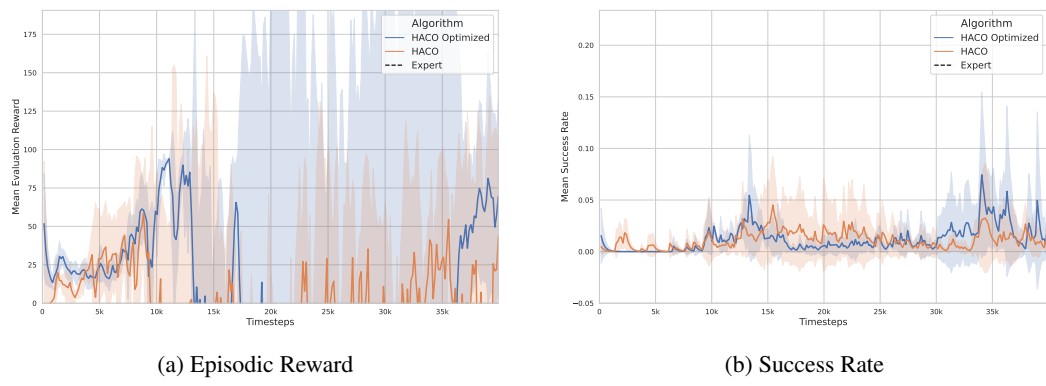

(a) Episodic Reward

(b) Success Rate

Figure 8: Generalizability on HACO. The enhanced **HACO+** (labeled 'HACO Optimized') shows faster and more stable learning than the original HACO.

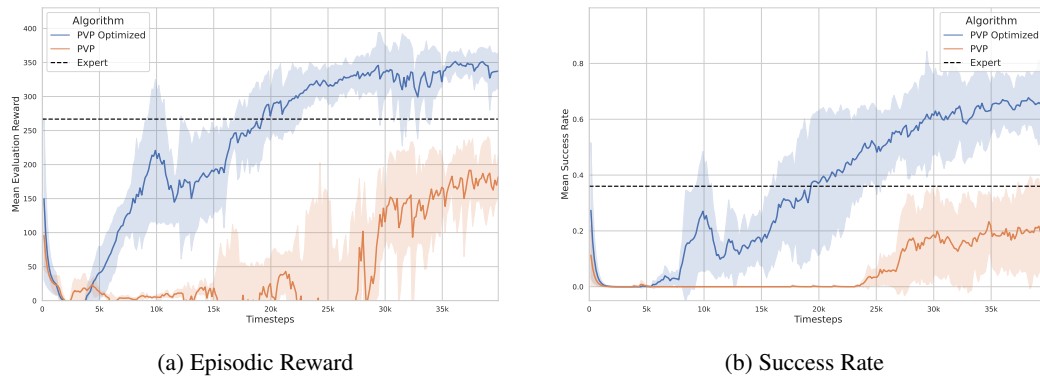

(a) Episodic Reward            (b) Success Rate

Figure 9: Generalizability on PVP. The enhanced **PVP+** (labeled 'PVP Optimized'), which is equivalent to our full VIQS model, dramatically outperforms the original PVP.

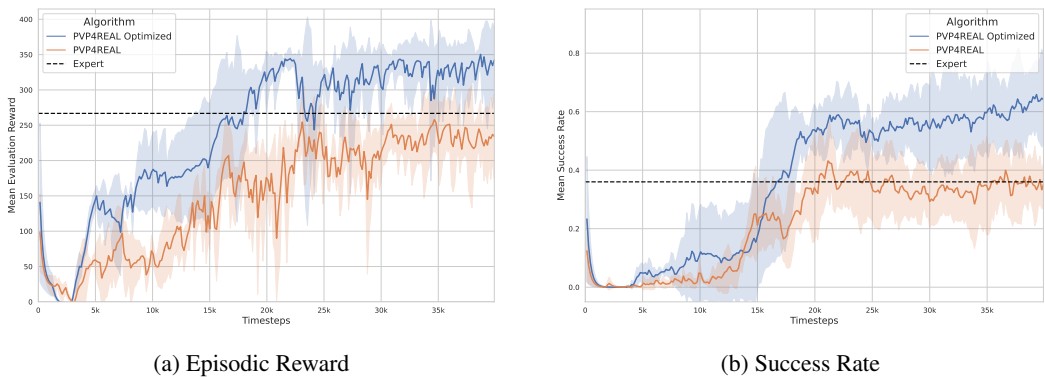

(a) Episodic Reward            (b) Success Rate

Figure 10: Generalizability on PVP4REAL. The enhanced **PVP4REAL+** (labeled 'PVP4REAL Optimized') achieves higher performance with better stability than the original.

# D ANALYSIS OF CORE MECHANISMS AND HYPERPARAMETERS

## D.1 ANALYSIS OF THE DISCRIMINATOR'S TRAINING DYNAMICS

A key design choice in VIQS is to train the discriminator for a fixed number of initial steps and then freeze its weights for the remainder of the agent's training. The rationale is to establish a *stable quality assessor* that provides a consistent learning signal, rather than a "moving target" that would complicate the critic's convergence.

To validate this choice, we plot the discriminator's accuracy and loss during the agent's training process (Figure 11). The results are shown across all three expert quality tiers and multiple random seeds. As depicted, in most runs, the discriminator's loss rapidly decreases and its accuracy quickly climbs to over 95% within the first 5,000 training steps. After this initial convergence, both metrics remain remarkably stable.

A notable case is the `LowExpert_seed600` run, where significant learning begins later, around the 10,000-step mark. **This is not a failure of the discriminator but rather a direct consequence of our data-efficient intervention mechanism.** The discriminator is trained exclusively on intervention data ($I_t = 1$). In this specific run, very few interventions were triggered during the early stages, meaning there was insufficient data for the discriminator to learn from. Once a critical mass of intervention data was collected, it began learning just as rapidly as in the other runs. This behavior underscores that the discriminator's training is driven solely by, and is a direct function of, the sparse human guidance signals.

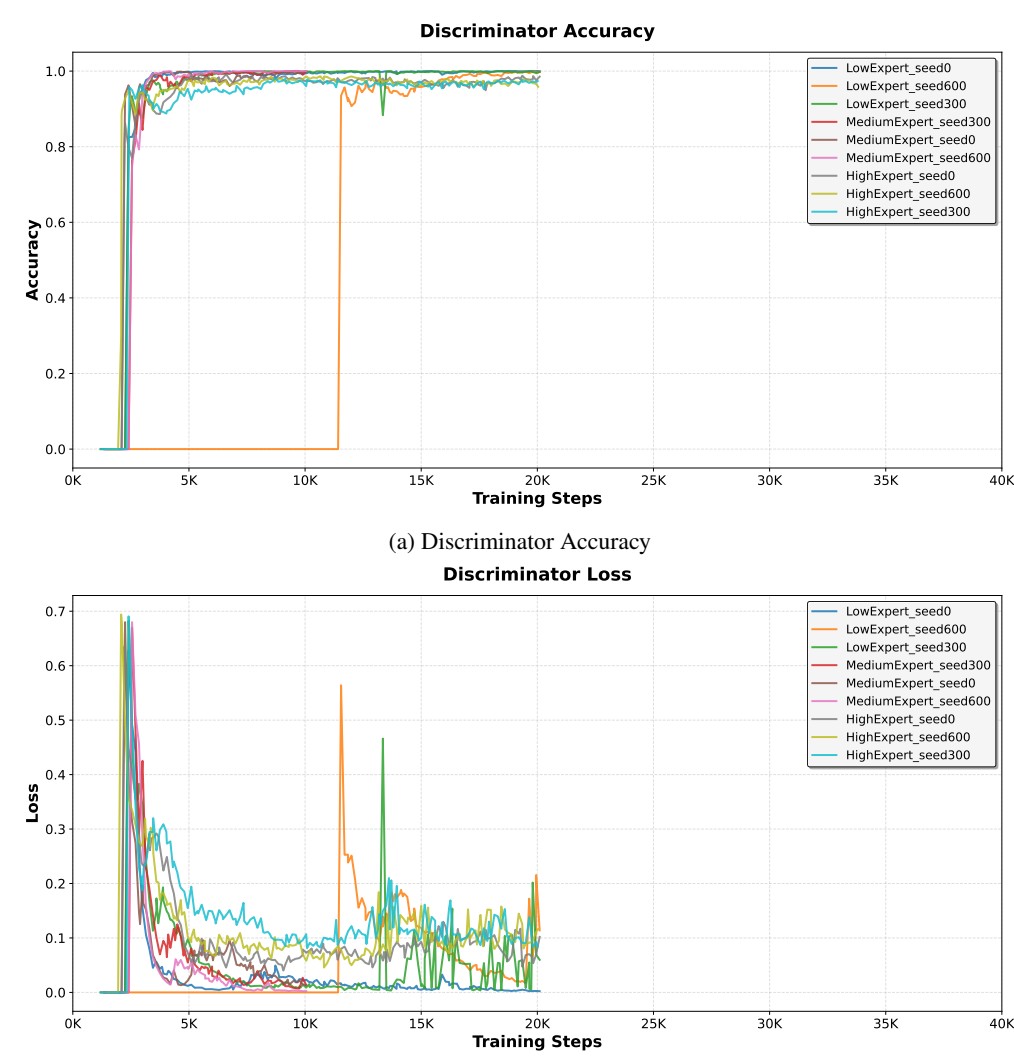

(a) Discriminator Accuracy

(b) Discriminator Loss

Figure 11: **Discriminator Training Dynamics.** Across all expert levels and random seeds, the discriminator's accuracy (a) rapidly converges to near-perfect, and its loss (b) stabilizes at a low value within the first 5,000 steps. This validates our design choice to freeze the discriminator early to create a stable quality assessor.

This empirical result strongly supports our design. It demonstrates that the discriminator can very efficiently learn to distinguish between the expert's and the novice agent's actions once data is available. Freezing the discriminator after its initial convergence thus provides a computationally efficient and stable mechanism for grounding the agent's reward-free learning process.

### D.2 SENSITIVITY ANALYSIS OF THE INTERVENTION THRESHOLD $\tau_Q$

We analyze the sensitivity of the intervention threshold $\tau_Q$, which balances agent autonomy against expert guidance. Experiments were conducted with the Medium-Quality expert (return $\approx 270$). Figure 12 demonstrates that VIQS is robust to the choice of $\tau_Q$ and that this hyperparameter can be set in a principled manner.

**Performance Robustness (Figure 12a):** The agent's final performance is robust across a broad range of $\tau_Q$ values from $[0.75, 1.25]$, where it consistently surpasses the expert baseline. This shows that VIQS does not require meticulous tuning. Thresholds that are too low ($\tau_Q = 0.5$) or too high

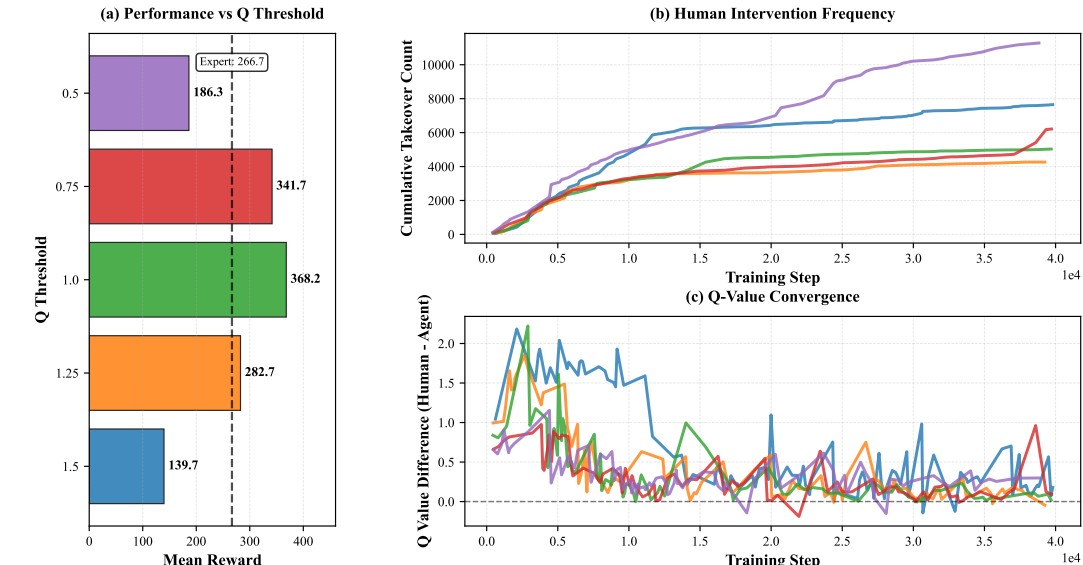

Figure 12: **Sensitivity analysis of the intervention threshold** $\tau_Q$ **with a Medium-Quality expert.**
(a) Agent performance is robust across a wide range of $\tau_Q$ values, consistently surpassing the expert
(dashed line). (b) The intervention cost decreases as $\tau_Q$ increases, showing a clear trade-off. (c) The
Q-value difference converges to zero, demonstrating successful learning and providing a principled
way to set $\tau_Q$.

($\tau_Q = 1.5$) degrade performance due to excessive intervention or insufficient correction, respectively.

**Complex Intervention Dynamics (Figure 12b):** This panel reveals a nuanced relationship between
$\tau_Q$ and intervention cost. Within the optimal range ($[0.75, 1.25]$), a lower $\tau_Q$ correctly leads to more
frequent interventions. **Notably, the curve for $\tau_Q = 1.5$ is steeper than for $\tau_Q = 1.25$ or $\tau_Q = 1.0$,
indicating a higher cumulative intervention count.** This is not contradictory. A threshold set too
high allows the agent's policy to degrade significantly. Consequently, while the instantaneous prob-
ability of intervention is low, the agent remains in a perpetual state of poor performance, requiring
more cumulative corrections over the long term. This highlights the importance of providing timely
guidance.

**Learning Dynamics and Principled Selection (Figure 12c):** This panel plots the Q-value differ-
ence from the expert's perspective, $Q(s_t, a_t^E) - Q(s_t, a_t^L)$, which triggers interventions. The differ-
ence starts at a large positive value (approx. +1.0) and converges toward zero as training progresses.
This directly evidences that the agent successfully internalizes the expert's guidance, aligning its
value estimates with the expert's. Crucially, the magnitude of this initial gap ($\approx 1.0$) corresponds
perfectly to the optimal threshold ($\tau_Q = 1.0$) found in Panel (a). This provides a powerful and
practical heuristic: $\tau_Q$ **can be effectively estimated by measuring this initial Q-value gap, thus
avoiding extensive hyperparameter search.**

# E    EMPIRICAL VALIDATION OF LEARNING STABILITY

To empirically address the reviewer's query regarding the stability and convergence of our learning
framework, we tracked the evolution of the Q-function throughout the training process. The analysis
was conducted under the guidance of experts with varying quality levels (High, Medium, and Low).
We present the results in Figure 13, which plots the average Q-values for two distinct action types:
the action executed in the environment (Behavioral Action) and the action proposed by the agent's
policy (Agent Action). The empirical results presented in Figure 13 provide a clear and direct answer
to the question of learning stability: **Primary Finding: Universal Stability and Convergence.**

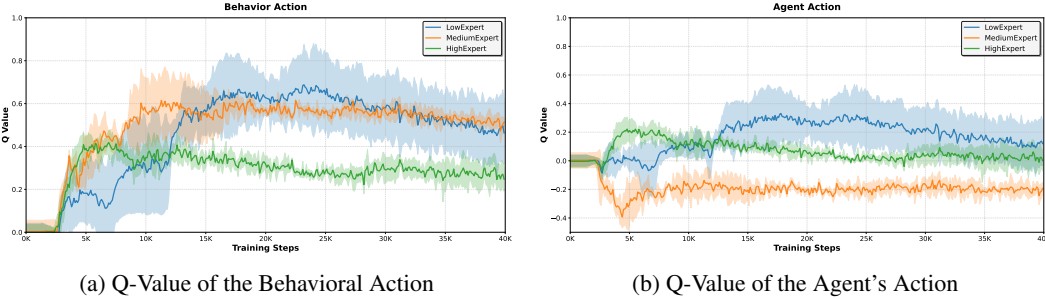

(a) Q-Value of the Behavioral Action          (b) Q-Value of the Agent's Action

Figure 13: Evolution of average Q-values under guidance from experts of different qualities, averaged over three random seeds. (a) shows the value of the executed actions. (b) shows the value of the agent's proposed actions.

The central observation across all experimental conditions is that the Q-value estimates remain bounded and exhibit clear convergence. Regardless of the expert's quality—be it High, Medium, or Low—and across both behavioral and agent-proposed actions, we observe no signs of divergence or instability. After an initial phase of adaptation, all curves flatten and converge to a steady-state value. This provides strong empirical evidence that our proposed method, VIQS, ensures a stable and well-behaved learning process, successfully avoiding the Q-value explosion or collapse that can plague similar offline or reward-free reinforcement learning algorithms.

# F  VALIDATION ON THE LUNARLANDERCONTINUOUS-V2 BENCHMARK

To demonstrate the generalizability of VIQS beyond the domain of autonomous driving, we evaluated it on the widely-used `LunarLanderContinuous-v2` classic control environment from OpenAI Gym (Brockman et al., 2016). This benchmark is a staple in the field, frequently employed to validate modern reinforcement and imitation learning algorithms. Its well-understood dynamics and continuous action space make it an ideal testbed for assessing sample efficiency and the ability to surpass suboptimal teachers, as demonstrated by its use in recent works such as Shah et al. (2025) and Zhu et al. (2025).

**Experimental Setup.**    Following a similar protocol to our main experiments, we utilized a pre-trained SAC agent as the simulated expert teacher. This expert is intentionally suboptimal, achieving an average evaluation reward of 129.7. All HIL algorithms, including VIQS, PVP, and PVP4REAL, were trained for a total of 50,000 environment steps.

Table 9: Performance comparison on `LunarLanderContinuous-v2`. All methods were trained for 50k steps with guidance from a suboptimal expert (Return ≈ 130). VIQS successfully surpasses the teacher and requires the fewest interventions.

| Expert Quality | Method | Training | Testing |
|---|---|---|---|
| | | Human Data Usage ($\downarrow$) | Episodic Return ($\uparrow$) |
| *Suboptimal Expert (Return ≈ 130)* | | | |
| | SAC / Expert Policy | - | 129.7 |
| | PVP | 35.5K $\pm$ 2.4K (0.71) | 43.3 $\pm$ 67.5 |
| | PVP4REAL | 23.6K $\pm$ 1.1K (0.47) | 129.3 $\pm$ 30.1 |
| | **VIQS (Ours)** | **10.8K $\pm$ 0.4K (0.22)** | **150.4 $\pm$ 34.9** |

**Results and Analysis.**    The results, presented in Table 9 and Figure 14, not only confirm the superiority of VIQS but also clearly illustrate the critical challenge of overcoming the **teacher-quality ceiling** in HIL-RL. Our method achieves a final return of 150.4, successfully surpassing the suboptimal teacher while requiring only 10.8k interventions—less than half that of PVP4REAL.

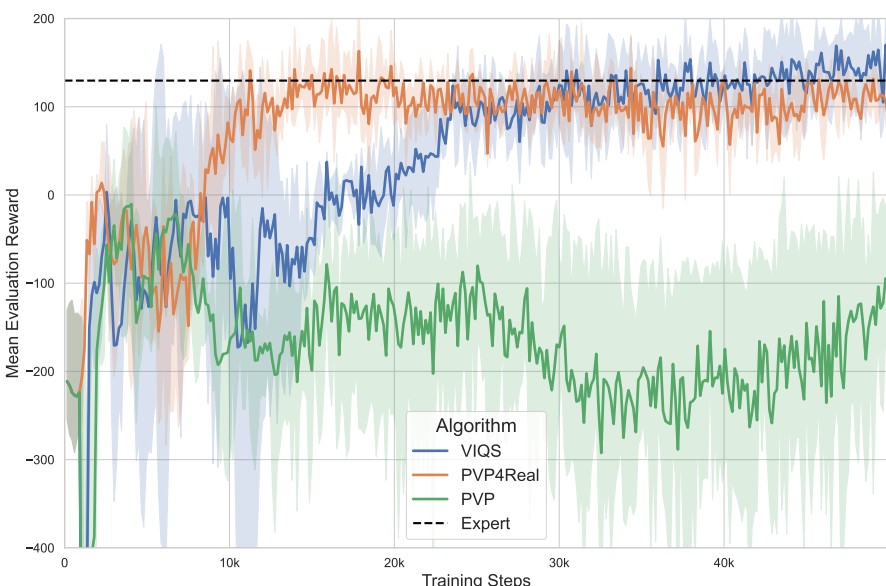

Figure 14: Learning curves on `LunarLanderContinuous-v2` over 50k training steps. VIQS demonstrates significantly faster learning and achieves a higher final reward, successfully surpassing its suboptimal expert teacher (performance indicated by the dashed line).

This success stands in stark contrast to the baseline methods. The conventional approach, PVP, completely fails; its performance collapses as frequent, unfiltered interventions from the suboptimal expert fundamentally corrupt the agent's value function, leading to a divergent policy. To combat this instability, PVP4REAL incorporates a BC loss into the actor's objective. While this prevents divergence, it does so by forcing the agent to mimic the teacher's actions, thereby erecting a hard **teacher-quality ceiling**. The agent is effectively shackled to the expert's suboptimal behavior, unable to explore and discover a superior policy.

Conversely, **VIQS is explicitly designed to shatter this ceiling**. Our approach decouples learning from direct policy imitation. Instead of constraining the actor's action space, we provide high-level, quality-aware guidance directly to the agent's value landscape. This liberates the agent from its teacher's policy constraints, empowering it to discover a superior solution. This ability to break the teacher-quality ceiling in a data-efficient manner demonstrates a robust and more practical paradigm for human-in-the-loop learning.

# G   IMPLEMENTATION AND ANALYSIS OF HIL-SERL BASELINE

## G.1   ADAPTATION FOR OUR FRAMEWORK

Adapting HIL-SERL from robotic manipulation to the class of problems addressed by our framework requires careful consideration of the differences in task structure, data quality, and reward signals. To maintain a fair and consistent comparison, we configured the HIL-SERL agent under the following conditions, which deviate from its typical optimal setup:

- **Sub-optimal Demonstration Buffer:** The core of HIL-SERL's sample efficiency relies on bootstrapping from a small set of high-quality expert demonstrations. However, a central challenge in many real-world applications, such as those targeted by our framework, is learning from imperfect data. To align with this premise, the demonstration buffer for HIL-SERL was populated using data from the same sub-optimal expert used for other methods. This inherently provides a lower-quality and biased starting point. Moreover, such data often contains inconsistent behavior patterns—where the expert makes different decisions in similar scenarios—which not only reduces learning efficiency but also introduces significant instability to the policy optimization process.

- **High-Risk Intervention Strategy:** HIL-SERL's methodology explicitly warns against providing long, sparse interventions that lead to task success, as this can cause overestimation of the value function and unstable training dynamics (Luo et al., 2024). However, this is precisely the nature of interventions in many long-horizon, high-stakes tasks. For instance, a human take-over to avert a critical failure often constitutes a long, sparse event that comprises a significant portion of the trajectory. When these interventions come from a sub-optimal expert, they may not even guarantee task success, further exacerbating the credit assignment problem. Specifically, such long takeovers make it nearly impossible for the algorithm to distinguish between critical corrective actions and superfluous or stylistic maneuvers from the expert, effectively diluting the learning signal and hindering effective policy updates.

- **Extremely Sparse Rewards from Simulation:** The HIL-SERL paper recommends training a dedicated binary reward classifier on collected data. In contrast, to maintain consistency across all baselines, we rely solely on the native binary success signal provided by the environment at the end of each episode. For HIL-SERL, which lacks a pre-trained value function to provide initial guidance, this extremely sparse, long-horizon reward makes learning from scratch exceptionally difficult. It forces the agent to rely almost entirely on the flawed initial demonstrations and interventions for a learning signal.

## G.2 PERFORMANCE ANALYSIS

The combination of these challenging factors—sub-optimal prior data, a risky intervention pattern, and extremely sparse rewards—exposes a key sensitivity of the HIL-SERL framework. Its mechanism is optimized to rapidly refine a policy around a high-quality "nominal trajectory" provided by expert demonstrations and corrections. When this core assumption of a near-optimal teacher is violated, its performance degrades substantially.

Therefore, its poor performance in our benchmark is an important finding. It does not refute the effectiveness of HIL-SERL in its intended domain. Rather, it highlights that its high-performance paradigm is fundamentally dependent on the availability of high-quality expert guidance. The learning mechanism is engineered to trust and efficiently leverage this guidance. When the expert is sub-optimal and provides inconsistent or flawed data, the very foundation of HIL-SERL's learning process is undermined. This underscores the need for methods like ours, which are explicitly designed to be robust to expert sub-optimality and sparse rewards, enabling them to learn effectively even when the teacher is imperfect.

