# OpenReview forum: "VIQS: Overcoming the Teacher Ceiling with Value-Guided Intervention and Quality-Aware Shaping"
_ICLR.cc/2026/Conference — Submitted to ICLR 2026_

### Official Review · Reviewer_geqJ · 2025-10-23

**Soundness:** 3
**Presentation:** 4
**Contribution:** 3
**Rating:** 6
**Confidence:** 3

**Summary:**

This paper addresses the problem of the "teacher-quality ceiling" in Human-in-the-Loop Reinforcement Learning (HIL-RL), where an agent's performance is bounded by the proficiency of the human expert. The authors propose the VIQS framework, which integrates two core mechanisms: a value-guided intervention trigger that compares the long-term value of the agent's action to an expert's reference action, and a quality-aware shaping mechanism that uses a discriminator to dynamically assess the quality of expert interventions. This operates in a reward-free setting, incorporating guidance through adaptively weighted losses for the critic. Experiments conducted in the MetaDrive simulator demonstrate that VIQS outperforms existing HIL methods in terms of performance and data efficiency.

**Strengths:**

1. The paper explicitly formalizes the "teacher-quality ceiling" as a limitation in existing HIL methods that rely on direct imitation or fixed proxy values, thereby clarifying and motivating the problem.
2. VIQS combines value-based triggering (avoiding stylistic mimicry) and discriminator-based quality shaping, generating adaptive proxy rewards. This allows the agent to learn "critically" from guidance, filtering suboptimal advice while absorbing high-quality demonstrations.
3. The paper is well-written and easy to follow. The authors effectively decompose the problem into "When to intervene" and "How to learn," logically structuring the paper.

**Weaknesses:**

1. The discriminator Dψ(s, a) plays a central role in shaping, yet no examination is provided regarding failure cases, early-stage instability, or its impact when misjudging high-value actions. Can the authors provide more analysis, such as visualizing the distribution of proxy values ($y_t$), to demonstrate the discriminator's stability and judgment?

2. Key hyperparameters such as τ_Q and α are fixed without robustness studies (Eq. 1, Eq. 4). Although Table 7 lists defaults, there is no exploration of performance sensitivity or tuning guidelines, leaving practical reproducibility uncertain. Could the authors provide a sensitivity analysis for these critical hyperparameters to validate the robustness of the results and offer guidance on implementation?

3. The intervention mechanism fundamentally relies on having access to an expert's action-value function, $Q^E$. This is a very strong assumption in practical HIL-RL scenarios involving real human experts, who do not have an explicit, accessible Q-function. How sensitive is the method to the quality of $Q^E$, and what modifications could make the framework applicable to real human experts who lack an explicit value function?

**Questions:**

Please see Weaknesses.

---

> ### Author Response · Authors · 2025-11-21
>
> Thank you for your excellent summary of our work and for your highly positive assessment. Your insightful questions have guided us in making substantial improvements to the paper. We have revised our manuscript to address all of your concerns, with changes clearly highlighted in yellow.
>
> **1. Weakness: Lack of analysis on discriminator stability and impact.**
>
> This is an excellent suggestion. To address your concern, we have added new analyses.
> *   **Discriminator Stability:** **Appendix D.1 (Figure 11, Pages 22-23)** visualizes the discriminator's accuracy and loss, showing it converges extremely quickly and remains stable.
> *   **Q-Value Stability:** As the proxy value $y_t$ is a direct transform of the discriminator's output, its stability is implied. Furthermore, our new **Appendix E (Figure 13, Page 25)** shows that the critic's Q-values also converge stably, providing comprehensive evidence for our framework's stability.
>
> **2. Weakness: Lack of robustness studies for key hyperparameters.**
>
> Thank you for highlighting this. We have added a comprehensive sensitivity study for `τ_Q` in a new **Appendix D.2 (Figure 12, Pages 23-24)**. The analysis shows that VIQS's performance is robust and reveals a principled heuristic for its tuning. We kept `α=1.0` as it is a standard implementation choice and not a sensitive parameter, allowing us to focus the analysis on our novel hyperparameters.
>
> **3. Weakness: Reliance on an expert's action-value function, $Q^E$, and its sensitivity.**
>
> This is a crucial point that touches on both the practical applicability of our method and its robustness. We address it from two complementary angles:
>
> *   **Practicality and Bridging to Real Human Experts:** We agree that assuming direct access to an explicit $Q^E$ is a key consideration for real-world deployment. Our "agent-as-expert" setup is instrumental for conducting **controlled and reproducible experiments**—a key strength allowing us to isolate the scientific question of how an agent can surpass an imperfect teacher. However, we have designed VIQS with real-world application in mind. In the newly expanded **"Limitations and Future Work" section (Page 10, highlighted)**, we detail a clear roadmap toward this goal, leveraging our framework's inherent modularity:
>     1.  **Decoupling Enables Flexibility:** Because our design inherently **decouples the intervention trigger from the quality-aware learning mechanism**, the trigger is plug-and-play. It can be replaced with $Q^E$-free signals, such as a **human-pressed intervention button**, without altering the core learning process. Our quality-shaping module would still learn critically from these interventions.
>     2.  **Learning a Proxy $Q^E$:** A viable pathway exists to implement our framework without a pre-trained expert model. Inspired by advances in preference-based learning (e.g., RLHF), a **proxy $Q^E$ can be effectively learned** from a pre-collected dataset of human judgments (e.g., pairwise comparisons). This learned model could then serve as the reference value function within VIQS.
>
> *   **Sensitivity to the Quality of $Q^E$:** Your question about sensitivity is, in fact, **the central scientific question our main experiments are designed to answer**. The "agent-as-expert" paradigm allows us to precisely control for this variable. Therefore, our main experimental setup using experts of **High, Medium, and Low** quality (**Section 6.2, Table 1, Page 8**) *is* precisely the sensitivity analysis you are suggesting. The results provide a clear and strong answer:
>     *   VIQS is **highly robust** to the quality of the guiding $Q^E$. It not only learns effectively from all three quality tiers but, more importantly, **consistently surpasses its guiding expert in every single case**. This demonstrates that the performance of VIQS is not brittle and gracefully adapts to a wide spectrum of expert proficiency, validating the core design of our method.
>
> We are confident that these additions have made our paper stronger. Thank you again for your valuable feedback.

---

> > ### Comment · Reviewer_geqJ · 2025-11-25
> >
> > We thank the authors for their diligent revisions, which have strengthened the paper through the addition of empirical analysis (e.g., discriminator stability, hyperparameter sensitivity) and a more thorough discussion of limitations. These enhancements are appreciated and directly address several of our initial concerns.
> >
> > However, our fundamental reservation regarding the core assumption of an accessible expert Q-function ($Q^E$) remains. While the proposed pathways for real-world application are noted, the method's validation within an idealized "agent-as-expert" paradigm still presents a significant gap to practical deployment with human teachers. Therefore, while we acknowledge the paper's conceptual contribution, we maintain our original rating (6) due to this persisting limitation.

---

> ### Author Response · Authors · 2025-11-25
>
> Thank you for your thoughtful follow-up and for acknowledging the value of our revisions. We are glad that the new empirical analyses and discussions have addressed many of your initial concerns.
>
> We sincerely appreciate you clearly articulating your fundamental reservation regarding the $Q^E$ assumption. Your comment perfectly captures the central trade-off we navigated in this work: the path to immediate practical deployment versus the scientific rigor required to isolate and validate a new learning principle.
>
> You rightly identify our work's primary strength as a conceptual contribution, and we fully embrace this characterization. Our main goal was to prove, in a controlled and reproducible manner, that it is possible to design a mechanism that systematically breaks the teacher-quality ceiling. The "agent-as-expert" paradigm, while creating the "gap" you noted, was the necessary scientific tool to achieve this. It allowed us to precisely control for teacher quality and provide clear, empirical evidence that our quality-aware shaping mechanism enables an agent to learn 'critically' rather than to imitate 'blindly'.
>
> We believe this foundational proof-of-concept serves as a valuable building block for the HIL-RL community. By providing empirical evidence that the ceiling can be broken by design in a controlled setting, our work aims to encourage and inform future research focused on bridging that very gap to practical human-in-the-loop systems.
>
> Your critical perspective has been immensely valuable. It has pushed us to not only strengthen the paper but also to more clearly define its contribution and its boundaries. We are very grateful for your insightful engagement.
>
> Sincerely,
>
> The Authors

---

### Official Review · Reviewer_L4Uj · 2025-10-28

**Soundness:** 3
**Presentation:** 3
**Contribution:** 3
**Rating:** 6
**Confidence:** 4

**Summary:**

The manuscript presents a novel framework that addresses the imperfect human decisions in the setting of human-in-the-loop reinforcement learning. The motivation is clear, the method is easy to follow, and the experiment results are promising. Nevertheless, there are a few questions concerning the model design that should be clarified more.

**Strengths:**

1. Clear motivation
2. Structured model description
3. Extensive and promising experimental results

**Weaknesses:**

1. Some unclear model details and settings.
2. Some experimental results conflict with the description of previous works.

**Questions:**

1. How is the value of the threshold $\tau_Q$ in equation (1) determined? I only saw a value in Table 7 in the Appendix, but there is no further analysis concerning this threshold.

2. There is a small gap between the two components. In the first component, only the ``high-value'' expert actions will trigger the learning. But in the second component, there is a further discriminator trying to filter out high- and low-quality advice. I am not questioning this design, just wondering whether the authors have tried to ensemble the two components together. For example, what if we only let the high-quality advice trigger the learning?

3. In line 214, the discriminator's parameters are frozen after a few steps. Given that the discriminator is trained with the agent's policy from scratch concurrently, how can the authors guarantee that the discriminator is well-trained after a fixed number of steps? Is it possible that the agent's policy learns slowly in the beginning and hence the discriminator learns a loose discrimination threshold?

4. In the introduction part, the authors claimed that existing works may perpetuate the teacher-quality ceiling because of blind trust, e.g., PVP and PVP4REAL, but it seems that, from Table 1, these two methods still outperform the baseline solutions purely using human expert policies. Compared with these two baselines, does VIQS provide other essential advantages?

**Details Of Ethics Concerns:**

N.A.

---

> ### Author Response · Authors · 2025-11-21
>
> Thank you for your constructive review. Your thoughtful questions have been instrumental in helping us strengthen the paper. We have carefully revised the manuscript to address all your points, with new text and figures highlighted for your convenience.
>
> **1. Question: How is the value of the threshold `τ_Q` determined?**
>
> This is an important practical question. We conducted a full sensitivity analysis for `τ_Q`, with detailed results in the new **Appendix D.2 (Figure 12, Pages 23-24)**. The analysis demonstrates that performance is robust across a wide range of `τ_Q` values and reveals a principled heuristic for its tuning, enhancing reproducibility.
>
> **2. Question: The gap between the two components (trigger and discriminator).**
>
> This is a very insightful question that gets to the heart of our design philosophy. The decoupling of the trigger and the shaper is a **deliberate and critical design choice**.
>
> To illustrate why, let us consider two "simpler" alternatives:
>
> *   **Alternative 1: Use the expert's Q-function ($Q^E$) directly for learning.** The primary issue is that the expert's value function is noisy and imperfect. Using it as a direct learning signal would force the agent to mimic the expert's suboptimal value landscape, effectively re-introducing the very "teacher-quality ceiling" we aim to break. Our design avoids this by using $Q^E$ *only* to trigger an intervention, while letting the more robust discriminator handle the actual learning signal.
>
> *   **Alternative 2: Use the discriminator to decide when to intervene.** The problem here is that the discriminator is trained to distinguish the agent's actions from the expert's. A trigger based on its output would be sensitive to any *stylistic* deviation, even if that deviation is strategically sound or even superior. This would degenerate the algorithm into an imitation learning framework.
>
> **Crucially, our existing ablation study empirically proves the failure of the second alternative.** The **"Action-based Trigger"** variant in **Table 3(Page 10)** suffers a catastrophic performance collapse (1.3% success rate). This provides strong empirical evidence that strategic intervention decisions (our value trigger) and quality assessment for learning (our discriminator) *must* be decoupled to achieve our goal.
>
> **3. Question: Guaranteeing the discriminator is well-trained when frozen early.**
>
>  This is a critical question of empirical validation. We added **Appendix D.1 (Figure 11, Pages 22-23)**, which visualizes the discriminator's training dynamics. The analysis shows that its accuracy surges to >95% very early in training, demonstrating that it learns a reliable and stable "expert-likeness" concept from sparse data, justifying our freezing strategy.
>
> **4. Question: Regarding PVP/PVP4REAL surpassing the expert and the advantage of VIQS.**
>
> This is an astute observation. The essential advantage of VIQS is that its ability to surpass the expert is **systematic and by design**. Unlike "blind trust" methods like PVP, VIQS employs "critical learning" via a discriminator. This leads to **consistent ceiling breakthroughs** across all expert tiers, the **discovery of novel and superior solutions** (**Figure 3 (Page 9)**), and **unmatched guidance efficiency** (2.4x to 5x fewer interventions), advantages we have now clarified in **Section 6.2 (Page 8)**.
>
> Thank you for prompting us to position our contribution more precisely.

---

> > ### Comment · Reviewer_L4Uj · 2025-11-25
> >
> > The above responses have addressed most of my concerns. Based on the overall evaluation and other reviewers' comments, I would maintain my current ratings.

---

> ### Author Response · Authors · 2025-11-25
>
> Thank you for your prompt feedback and for confirming that our responses have clarified most of your concerns.
>
> We hope that our extensive revisions—particularly the new "Limitations and Future Work" section and the added justifications—have successfully framed our assumption of an accessible expert Q-function not as a fundamental flaw, but as a deliberate choice for a controlled scientific study, while also providing a clear and viable roadmap for future work.
>
> We are very grateful for your time and your constructive engagement throughout this process. Your feedback has been invaluable in helping us significantly strengthen our manuscript.
>
> Sincerely,
>
> The Authors

---

### Official Review · Reviewer_sst8 · 2025-10-28

**Soundness:** 2
**Presentation:** 3
**Contribution:** 2
**Rating:** 4
**Confidence:** 4

**Summary:**

This paper proposes a method, VIQS, to address HIL-RL in a reward-free setting. The paper has two main contributions: (1) proposing when to conduct active intervention; (2) how to train the Value function with expert data and without external reward. The paper then demonstrates in the MetaDrive environment that their algorithm, after receiving a certain amount of expert data, achieves higher efficiency than traditional online RL established from-scratch via pure exploration .

**Strengths:**

### 1. High writing quality:
The paper is well-written with a clear and rigorous logical structure, making it easy for readers to understand.

### 2. Reasonable intervention mechanism:
One of the paper's key contributions is proposing the use of a value-based comparison ($Q^E(s,a^E) - Q^E(s,a^L)$) rather than an action-based comparison to trigger intervention. This is a reasonable choice because the Q-value (policy value) carries more task-relevant information than the action itself (policy appearance), which allows the intervention to focus on strategy rather than style.

### 3. Novel reward-free Q-learning:
In a "reward-free" setting, using the discriminator's (GAN) output $D(s,a)$ to construct a bounded proxy value target ($y=2D-1$)—and using this as an anchor ($\mathcal{L}_{PV}$) to shape the Q-function—is a relatively novel design.

**Weaknesses:**

### 1. Insufficient experimental validation:
The method is only tested in a single environment, METADRIVE, which is insufficient to demonstrate its generalizability in different HIL scenarios (e.g., robotic manipulation). Furthermore, the comparison of HIL baselines is limited; other methods in this field (such as HIL-SERL) should be considered for inclusion.

### 2. Lack of sensitivity analysis for key parameters:
The method has two hyperparameters: the value intervention threshold $\tau_Q$ and the training stop point for the discriminator (GAN). The paper does not provide ablation or sensitivity experiments for these two parameters, which raises questions about the method's robustness and the difficulty of tuning.

### 3. Inadequate explanation of core mechanisms:
The paper's explanation and justification for several key mechanisms (GAN learning and Q value) are somewhat vague (see detailed questions section).

**Questions:**

1. Could the authors discuss a broader range of HIL-RL baselines? The comparison is limited, and including methods with different mechanisms (e.g., HIL-SERL) would provide a more comprehensive benchmark.


2. Could the method's robustness be validated on more experimental environments beyond METADRIVE? For example, robotic manipulation tasks or other virtual environments would strengthen the claims of generalizability.


3. The method seems potentially sensitive to key hyperparameters, particularly the value intervention threshold $\tau_Q$. Could the authors provide a parameter sensitivity analysis for this?


4. Regarding the claim that "extending this knowledge across the state-space"(Line308, Page 6), according to the update(Eq. 6), a "value anchor" at time $t$ propagates backward in time to states $t-1$, $t-2$, etc. This suggests knowledge is propagated to preceding states, not necessarily the entire state-space. Could the authors clarify this and provide Q-value curves over time to demonstrate convergence and stability?


5. Regarding the GAN training: At what point is the discriminator training stopped? The paper states it's frozen after "a fixed number of training steps", but the policy $\pi_L$ continues to update. How can a static $D$ accurately evaluate an evolving $\pi_L$? This affects the quality of the Q function. Can I interpret this to mean that when the GAN network stops training, it has already completely learned to (distinguish) the expert's actions?

---

> ### Author Response · Authors · 2025-11-21
>
> Thank you for your detailed review and your positive feedback on our paper's writing quality, intervention mechanism, and novel Q-learning design. Your insightful questions have pushed us to provide deeper validation and clearer explanations. We have revised our manuscript accordingly, with all changes highlighted in yellow.
>
> **1. Weakness: Insufficient experimental validation (single environment, limited baselines).**
>
> We appreciate these excellent suggestions and have addressed both points.
>
> *   **Broader Baselines (HIL-SERL):** In direct response to your suggestion, we have **added HIL-SERL (Luo et al., 2024), a state-of-the-art method in robotic manipulation, to our baselines.** We have integrated it into our **RELATED WORK (Section 2, Page 3)** and included a full set of results in **Table 1 (Page 8)**. Furthermore, a **detailed analysis is provided in the new Appendix G (Pages 26-27)**, explaining that HIL-SERL's design is optimized for near-optimal guidance, which causes it to struggle in our "learning from imperfect experts" setup. This highlights a key difference in problem settings and reinforces the novelty of our approach.
>
> *   **More Environments:** To validate the generalizability of VIQS, we have conducted new experiments on the **LunarLanderContinuous-v2** classic control benchmark. The results are now in a new **Appendix F (Pages 25-26)**. This benchmark's utility is well-established in recent literature (e.g., Shah et al., 2025; Zhu et al., 2025). The results (**Table 9, Figure 14(Pages 25-26)**) are highly consistent with our findings in MetaDrive: **VIQS successfully surpasses a suboptimal expert, while competitive baselines either fail to learn or are trapped by the teacher-quality ceiling.** This validation in a distinct domain demonstrates that our core principles are broadly applicable, significantly strengthening our claims.
>
> **2. Weakness: Lack of sensitivity analysis for key hyperparameters.**
>
> These are excellent questions. We have conducted new experiments and added new appendices to address them.
>
> *   **Sensitivity Analysis for $τ_Q$:** We have performed a detailed sensitivity analysis, presented in the new **Appendix D.2 (Figure 12, Pages 23-24)**. The results show that VIQS is robust across a wide range of $τ_Q$ values and reveal a principled heuristic for its tuning.
>
> *   **Discriminator Training and Rationale for Freezing:** We freeze the discriminator after an initial training phase to establish a *stable quality assessor*. As the agent's policy improves, its actions score progressively higher against this fixed "quality ruler," providing a consistent learning signal. To prove this empirically, we have added **Appendix D.1 (Figure 11, Pages 22-23)**, which shows the discriminator converges very rapidly and remains stable.
>
> **3. Weakness: Inadequate explanation of core mechanisms.**
>
> Thank you for this sharp question. We have revised the text in **Section 5 (Page 7)** to more accurately describe the mechanism as a two-stage process of temporal propagation and spatial generalization. To empirically demonstrate stability, we have added a new **Appendix E with Figure 13 (Page 25)**, which plots the Q-function's convergence and shows that Q-values remain bounded and stable.
>
> We are grateful for your constructive criticism, which has significantly improved our paper.

---

> > ### Author Response · Authors · 2025-11-29
> > **Follow-up on Weakness 1: Direct Results from New Experimental Validations**
> >
> > Thank you for your insightful feedback. In response to your comments on **Weakness 1 (Experimental Validation)**, we have prepared a comprehensive analysis comparing our method, VIQS, against the HIL-SERL baseline. This analysis spans **all three expert quality tiers** and includes results from a separate benchmark to demonstrate generalizability.
> >
> > **1. Validating Our Problem Space: Consistent Failure vs. Consistent Success**
> >
> > Our main claim is that existing methods, while powerful, can be brittle when their core assumption of high-quality expert guidance is violated. The following results from **Table 1 (Page 8)** provide stark empirical validation for this claim:
> >
> > | Expert Quality | Method | Human Data Usage (↓) | Episodic Return (↑) | Safety Cost (↓) | Success Rate (%) (↑) |
> > | :--- | :--- | :--- | :--- | :--- | :--- |
> > | **High** (≈350) | Expert Policy | - | 348.9 | 0.76 | 74.0 |
> > | | HIL-SERL | 28.1k ± 4.4k | 72.5 ± 100.6 | 18.5 ± 30.3 | **0.0 ± 0.0** |
> > | | **VIQS (Ours)** | **7.6k ± 1.4K** | **357.3 ± 20.7** | 2.1 ± 1.7 | **86.7 ± 6.4** |
> > | **Medium** (≈270)| Expert Policy | - | 266.7 | 0.7 | 36.0 |
> > | | HIL-SERL | 30.7k ± 1.6k | 75.5 ± 26.9 | 3.0 ± 3.5 | **0.0 ± 0.0** |
> > | | **VIQS (Ours)** | **4.1k ± 0.3k** | **334.7 ± 13.5** | **0.6 ± 0.5** | **70.7 ± 10.3** |
> > | **Low** (≈190) | Expert Policy | - | 186.3 | 0.76 | 16.0 |
> > | | HIL-SERL | 31.4k ± 0.4k | 63.5 ± 16.5 | 7.8 ± 11.6 | **0.0 ± 0.0** |
> > | | **VIQS (Ours)** | **4.0k ± 0.9K** | 265.4 ± 5.1 | **0.6 ± 0.1** | 38.7 ± 4.6 |
> >
> > This direct comparison reveals a critical pattern:
> >
> > *   **HIL-SERL's performance consistently collapses to 0% success, regardless of expert quality.** This is an expected but crucial finding. As detailed in our **Appendix G (Pages 26-27)**, this failure stems from a fundamental mismatch between HIL-SERL's design and our problem setting:
> >     1.  It was bootstrapped from **sub-optimal and inconsistent demonstrations**, rather than the high-quality priors it is engineered for.
> >     2.  It was subjected to **long, sparse human interventions**—a pattern its methodology explicitly cautions against due to training instability.
> >     3.  Crucially, it was forced to learn from the environment's final, binary success signal. This signal becomes an **unreliable and misleading learning target** in our setting because the **sub-optimal expert itself has a low success rate**. This creates a credit assignment catastrophe: the algorithm cannot learn which expert actions are valuable when those actions rarely lead to the very success signal it's supposed to chase.
> >
> >     Therefore, its failure is not a critique of HIL-SERL in its intended domain. Rather, it is a powerful empirical validation that a critical gap exists—one for which our robust method, VIQS, is specifically designed.
> >
> > *   **VIQS demonstrates remarkable robustness, consistently breaking the teacher-quality ceiling across all tiers.** It substantially outperforms its guiding expert in both success rate and return, while being vastly more data-efficient (requiring **4-7x fewer interventions** than HIL-SERL) and maintaining excellent safety.
> >
> > **2. Generalizability: The Same Winning Pattern Holds on LunarLander**
> >
> > To confirm this principle of robust, data-efficient learning is general, we present the full results from `LunarLanderContinuous-v2` (**Table 9, Page 25**), trained with a suboptimal expert:
> >
> > | Method | Human Data Usage (↓) | Episodic Return (↑) |
> > | :--- | :--- | :--- |
> > | *Suboptimal Expert (Return ≈ 130)* | - | 129.7 |
> > | PVP4REAL | 23.6k ± 1.1k | 129.3 ± 30.1 |
> > | **VIQS (Ours)** | **10.8k ± 0.4k** | **150.4 ± 34.9** |
> >
> > The data directly shown here reinforces the same narrative: VIQS is the **only method to significantly surpass its teacher's performance (+16%)** while also being dramatically more efficient, requiring **less than half the interventions** of its closest competitor.
> >
> > We trust that this comprehensive data and sharpened analysis now fully address your concerns, demonstrating the clear need for our method and its consistent, superior performance.

---

> > ### Author Response · Authors · 2025-11-29
> > **Follow-up on Weakness 2: Extensive Sensitivity Analysis and Principled Hyperparameter Selection**
> >
> > Thank you for your valuable feedback regarding the need for a sensitivity analysis. To comprehensively address this, we have added a **new appendix section (Appendix D, Pages 22-24)** dedicated entirely to analyzing our core mechanisms and learning stability.
> >
> > These analyses demonstrate not only that VIQS is robust, but also that its key hyperparameter can be set in a principled manner, avoiding costly tuning.
> >
> > Here is a summary of our key findings, with references to the figures you can find in the new PDF:
> >
> > **1. Stability of the Quality Assessor (Discriminator)**
> >
> > First, we analyzed our design choice of freezing the discriminator after a brief initial training phase.
> > *   **Finding:** As shown in **Figure 11 (Appendix D.1, Page 23)**, the discriminator demonstrates highly efficient learning. While most runs show convergence within the first 5,000 steps, some runs (e.g., `LowExpert_seed600`) begin training later. This is not a flaw, but rather an **expected and validating outcome of our data-efficient design.** The discriminator is trained *exclusively* on intervention data. In runs with very few early interventions, there is simply no data to learn from. Once a critical mass of intervention data is collected, the discriminator's accuracy rapidly climbs to over 95% and its loss stabilizes.
> > *   **Implication:** This proves that the discriminator is a highly effective and data-efficient classifier that learns rapidly *once data is available*. Freezing it after this quick convergence provides a **stable, consistent quality signal** for the agent. This validates our design choice as both computationally efficient and beneficial for stable learning.
> >
> > **2. Sensitivity and Principled Selection of the Intervention Threshold ($\tau_Q$)**
> >
> > This is the core of our sensitivity analysis, focused on the most critical hyperparameter, $\tau_Q$. The results in **Figure 12 (Appendix D.2, Page 24)** are threefold:
> >
> > *   **(a) Performance Robustness:** The agent's final performance is **highly robust** across a wide range of $\tau_Q$ values (from 0.75 to 1.25), consistently and significantly outperforming the expert. This demonstrates that VIQS is not brittle and does not require meticulous tuning to be effective.
> >
> > *   **(b) Understandable Intervention Dynamics:** The relationship between the threshold and human intervention cost is clear and intuitive. Setting the threshold too high ($\tau_Q=1.5$) reveals a crucial insight: while instantaneous interventions are rarer, the agent's policy degrades so much that it requires *more cumulative help* in the long run. This underscores the importance of timely guidance.
> >
> > *   **(c) Principled Selection to Avoid Tuning:** Most importantly, we show that this hyperparameter can be set **without extensive grid searching**. **Figure 12c** shows that the initial Q-value difference between the expert's and the agent's actions is approximately 1.0. This value corresponds *perfectly* with the optimal performance threshold found in our sweep ($\tau_Q=1.0$). This provides a powerful and practical heuristic: **one can simply measure this initial Q-value gap to set $\tau_Q$**, transforming it from a "magic number" into a parameter with a principled, data-driven basis.
> >
> >
> > In summary, these new analyses demonstrate that our method is not only robust to its key hyperparameter but also provides a principled heuristic for its selection, significantly reducing tuning overhead. We believe this comprehensively addresses your concerns and strengthens the paper's contribution.

---

> ### Author Response · Authors · 2025-11-29
> **Follow-up on Weakness 3: Clarification on Value Propagation and Empirical Proof of Stability**
>
> Thank you for this insightful question. It highlights a critical nuance in our learning mechanism, and you are absolutely correct in your observation: the standard TD-update rule, in isolation, primarily propagates value information **temporally along experienced trajectories**. Your point is well-taken, and our original phrasing "extending this knowledge across the state-space" was imprecise.
>
> To address this, we have made a two-part update to the paper:
>
> **1. Clarification of the Mechanism (Revised text on Page 6):**
>
> We have revised the text on Page 6 to more accurately describe how the value function is learned. The key insight, which we now state explicitly, is that generalization happens through a **synergy between two distinct processes**:
>
> - **Temporal Propagation (via TD Loss):** As you noted, this is the mechanism that backs up value from a state $s_t$ to its predecessors ($s_{t-1}$, etc.) along a specific path taken by the agent.
> - **Spatial Generalization (via Neural Network Approximator):** Crucially, because our critic is a neural network, it learns a continuous function over the state representation. An update for a single state-action pair also influences the Q-value estimates for _other, nearby states_ in the learned feature space.
>
> As our newly added text clarifies, it is the combination of these two effects that allows the trajectory-specific "value anchors" to inform the value estimates over a much wider portion of the state space, not just the exact trajectories visited.
>
> **2. Empirical Proof of Convergence and Stability (New Appendix E):**
>
> To provide the empirical validation you requested, we have added a new **Appendix E,Pages 24-25 ("Empirical Validation of Learning Stability")**. This appendix contains the very Q-value curves over time that you suggested we include.
>
> - As shown in **Figure 13(Page 25)** in this new appendix, the Q-values for both the agent's and the behavior policy's actions demonstrate **clear convergence and remain perfectly stable** across all experimental settings (High, Medium, and Low-quality experts).
> - This empirically confirms that our value learning process, driven by the synergy described above, is well-behaved and successfully avoids the value explosion or divergence that can be a concern in RL.
>
> We believe that this two-part response—a conceptual clarification in the main text and robust empirical evidence in the appendix—fully addresses your concerns. We are grateful for your feedback, as it has pushed us to make the underlying mechanics of our method significantly clearer for all readers.

---

### Official Review · Reviewer_fVz9 · 2025-10-31

**Soundness:** 3
**Presentation:** 3
**Contribution:** 2
**Rating:** 4
**Confidence:** 3

**Summary:**

This paper addresses the "teacher-quality ceiling" in HIL-RL. The authors propose VIQS, a reward-free framework featuring two components: value-guided intervention for intervention timing and quality-aware shaping for robust learning. Experiments on the MetaDrive benchmark demonstrate that VIQS outperforms existing HIL baselines and, notably, consistently surpasses the performance of its guiding expert.

**Strengths:**

1. The paper addresses a significant challenge in HIL-RL: enabling an agent to learn from an imperfect expert and eventually surpass the expert's performance. This capability is critical for real-world applications.

2. VIQS consistently outperforms its guiding expert across all quality levels, directly demonstrating its ability to break the teacher-quality ceiling.

**Weaknesses:**

1. The reliance on an expert Q-function is  a strong assumption. Value-guided Intervention requires access to $Q^{E}$ to evaluate the agent's actions. It's unclear if $Q^{E}$ can accurately judge the agent's new actions, which might be very different from the expert actions. Furthermore, a real human expert does not have a $Q^{E}$ network. This method seems designed for an agent-guiding-agent setup, not a human-guiding-agent setup. This limitation should be explicitly discussed.

2. The training data for the quality-aware discriminator appears to be extremely sparse. The quality-aware discriminator is trained only when an intervention happens and VIQS needs very few interventions. This could lead to unstable training or an unreliable discriminator. An analysis of the discriminator's training stability and sample efficiency is missing.

3. All experiments are conducted on the MetaDrive benchmark. It should be evaluated on other benchmarks to strengthen the empirical evidence.

4. The paper is missing clear definitions for several terms, such as $\delta_{a}$ in line 241 and $\pi'$ in Eq.(6).

**Questions:**

Please see the weakness part above.

---

> ### Author Response · Authors · 2025-11-21
>
> Thank you for your thorough review and insightful comments. We are grateful for your recognition of our work’s central contribution in addressing the "teacher-quality ceiling." Your feedback has been invaluable in helping us clarify our assumptions, strengthen our empirical validation, and better position our contributions. We have carefully revised the manuscript according to your suggestions, with all significant modifications highlighted in yellow for your convenience.
> Below, we provide a point-by-point response to the weaknesses you identified.
>
> **1. Weakness: The strong assumption of an expert Q-function ($Q^E$) and applicability to real human experts.**
>
> We wholeheartedly agree that assuming direct access to an explicit $Q^E$ does not directly map to a typical real-world human-in-the-loop setting. Our response to this crucial point is twofold:
>
> *   **Justification for Our "Agent-as-Expert" Paradigm:** Our work is situated within an effective research paradigm in HIL-RL (e.g., Xue et al., 2023; Xue et al., 2025) using an "agent-as-expert" setup. This approach is instrumental for conducting **controlled and reproducible experiments, which is a key strength of our work. It allows us to precisely vary the expert's quality and isolate the core scientific question: *How can an agent learn to systematically surpass an imperfect teacher?*** A core innovation of our framework is the value-guided trigger, which moves beyond the limitations of naive action-based triggers. By intervening based on long-term strategic value ($Q$-value) rather than stylistic action differences, our trigger more closely aligns with how a real human expert makes strategic decisions, while remaining fully controllable and reproducible in an experimental context.
>
> *   **Bridging to Real-World Application:** While our setup is justified for research, we recognize the importance of demonstrating its real-world viability. We have added a comprehensive discussion in the newly expanded **Limitations and Future Work" section (Page 10, highlighted)**. We now explicitly discuss this and propose concrete pathways towards real-world application:
>     1.  Because our framework's design inherently **decouples the intervention trigger from the quality-aware learning mechanism**, it allows for practical adaptations such as replacing the trigger with $Q^E$-free signals (e.g., a human-pressed button). Our quality-shaping module would still learn critically from the resulting intervention.
>     2.  **There exists a viable pathway to implement our framework without a pre-trained expert model. Inspired by advances in preference-based learning (e.g., RLHF), a proxy $Q^E$ can be effectively learned** from a pre-collected dataset of human judgments (e.g., pairwise comparisons). This learned model could then serve as the reference value function within the VIQS framework.
>
> **2. Weakness: Potentially sparse training data for the quality-aware discriminator.**
>
> This is an excellent and insightful point. To empirically validate the stability of our discriminator under the sparse regime it operates in, we have conducted a new analysis of its training dynamics.
>
> The results, presented in the new **Appendix D.1 (Figure 11, Pages 22-23)**, show that the discriminator's accuracy rapidly converges to >95% and its loss stabilizes very early in training. This demonstrates that it is highly sample-efficient and learns a stable decision boundary from a surprisingly small amount of intervention data, justifying our design choice to freeze it as a stable quality assessor.
>
> **3. Weakness: Experiments are limited to the MetaDrive benchmark.**
>
> We agree that testing on a wider range of benchmarks is crucial. To address this, we have taken two significant steps:
>
> 1.  **New Benchmark: LunarLander:** We have conducted new experiments on the classic control benchmark, **LunarLanderContinuous-v2**, adding the results to a new **Appendix F (Pages 25-26)**. This benchmark has been effectively used in recent, relevant works such as Shah et al. (2025) and Zhu et al. (2025). The results (**Table 9, Figure 14(Pages 25-26)**) are highly consistent with our findings in MetaDrive: VIQS successfully surpasses a suboptimal expert, while competitive baselines either fail to learn or are trapped by the teacher-quality ceiling. This validation in a distinct domain significantly strengthens our claims.
>
> 2.  **Broader Baseline from a Different Domain:** We have added **HIL-SERL**, a state-of-the-art HIL method from the robotics domain, to our baselines in MetaDrive (see **Table 1, Page 8**).
>
> **4. Weakness:  Missing definitions for terms.**
> Thank you for pointing this out. We have added clear definitions for ∆_a (Figure 2 caption, Page 5) and π' (following Equation 6, Page 6) in the revision.
>
> We are confident these revisions have substantially improved the clarity and rigor of our paper. We thank you again for your constructive feedback.

---

> > ### Author Response · Authors · 2025-11-29
> > **Follow-up on Weakness 2: Potentially sparse data for the discriminator**
> >
> > Thank you for raising this excellent and insightful point. You are correct to question the stability of the discriminator, given that it is trained only on sparse intervention data. A mechanism that relies on few interventions must indeed be highly sample-efficient.
> >
> > To empirically validate this, we conducted a completely new analysis of the discriminator's training dynamics throughout the learning process. The full results and discussion are now included in the new **Appendix D.1 (Figure 11, Pages 22-23)**.
> >
> > Our key findings are definitive and strongly support our design:
> >
> > *   **High Sample Efficiency:** The analysis shows that the discriminator's accuracy rapidly converges to over 95% and its loss stabilizes in the very early stages of training. This often happens with fewer than 5,000 environmental steps, which corresponds to only a small handful of intervention events.
> > *   **Validation of Design:** This demonstrates that the binary classification task we pose—distinguishing agent actions from expert interventions in specific states—is learned with surprising efficiency. In some runs where early interventions are extremely rare (e.g., `LowExpert_seed600`), the discriminator waits for a critical mass of data and then learns just as rapidly. This behavior is not a flaw; it is an expected and validating outcome of its data-driven nature.
> >
> > This empirical result confirms that the discriminator is not a point of failure but rather a robust and highly efficient component of our framework. It justifies our design choice to freeze the discriminator after its brief initial convergence, as it provides a stable and reliable "quality assessor" for the remainder of training without requiring further data or tuning.
> >
> > We are confident this new analysis provides the concrete evidence needed to address your concerns about the stability and sample efficiency of the discriminator.

---

> > ### Author Response · Authors · 2025-11-29
> > **Follow-up on Weakness 3: Experiments limited to the MetaDrive benchmark**
> >
> > Thank you for your feedback on the scope of our experimental validation. We agree completely that demonstrating the generality of our method beyond a single benchmark is crucial for establishing a strong contribution.
> >
> > To address this rigorously, we have undertaken **two significant expansions** to our empirical evaluation, which we believe substantially strengthen the paper.
> >
> > **1. New Benchmark Domain: LunarLanderContinuous-v2:**
> >
> > We have conducted a full suite of new experiments on the classic OpenAI Gym control benchmark, **LunarLanderContinuous-v2**. The complete results and analysis have been added to a **new Appendix F (Pages 25-26)**. We deliberately chose this domain because its dynamics and state space are fundamentally different from autonomous driving.
> >
> > The results are remarkably consistent with our findings in MetaDrive. To make our claims immediately verifiable, we present the core data from **Table 9 （Page 25）** directly below:
> >
> > | Method | Human Data Usage (↓) | Episodic Return (↑) |
> > | :--- | :--- | :--- |
> > | *Suboptimal Expert (Return ≈ 130)* | - | 129.7 |
> > | PVP | 35.5k ± 2.4k | 43.3 ± 67.5 |
> > | PVP4REAL | 23.6k ± 1.1k | 129.3 ± 30.1 |
> > | **VIQS (Ours)** | **10.8k ± 0.4k** | **150.4 ± 34.9** |
> >
> > This data paints a clear and compelling picture:
> >
> > *   **Surpassing the Teacher:** VIQS is the **only method to significantly surpass its suboptimal teacher (+16%)**, demonstrating its ability to break the teacher-quality ceiling in a completely different environment.
> > *   **Data Efficiency:** VIQS achieves this superior performance while being dramatically more efficient, requiring **less than half the interventions** of the next-best baseline (PVP4REAL).
> > *   **Baseline Limitations:** The performance of the baselines further validates our claims. PVP fails catastrophically, while PVP4REAL (an improved version of PVP) is fundamentally "shackled" to the teacher's performance, unable to discover a better policy.
> >
> > This successful validation in a distinct and widely-used domain significantly strengthens the generality of our claims.
> >
> > **2. Broader Baseline Comparison and Deeper Analysis in MetaDrive:**
> >
> > To further strengthen our primary results, we have also added **HIL-SERL**, a state-of-the-art HIL method from the robotics domain, to our baseline comparisons in MetaDrive (see revised **Table 1, Page 8**).
> >
> > While VIQS soundly outperforms HIL-SERL, it's the *reason* for HIL-SERL's poor performance that is most illuminating. We have dedicated a **new Appendix G (Pages 26-27) to a detailed analysis** of this result. In short, HIL-SERL's core design assumes access to high-quality demonstrations and short, corrective interventions. Our problem setting—featuring sub-optimal experts and long-horizon interventions—systematically violates its core assumptions, causing its performance to collapse. This finding is crucial: it does not refute HIL-SERL's effectiveness in its intended domain, but rather **provides powerful empirical evidence for the existence of the very problem gap that VIQS is specifically designed to fill.**
> >
> > With these two major additions, our paper now demonstrates the success and robustness of VIQS across multiple domains and against a wider set of baselines, all while providing deeper analysis that validates our core research motivation. We hope you'll agree that this significantly enhances the empirical evidence supporting our contributions.

---

### Author Response · Authors · 2025-11-21
**Summary of Revisions**

Dear Reviewers,

We extend our sincere gratitude to all of you for your time and invaluable feedback. Your insightful comments have been instrumental in helping us substantially improve our manuscript. In response to your suggestions, we have undertaken a significant revision, with all changes highlighted in yellow in the updated paper. The primary modifications include:

1.  **Strengthened Empirical Validation:**
    *   **New Benchmark (LunarLander):** To demonstrate the generalizability of our method, VIQS, we have conducted a full set of experiments on the `LunarLanderContinuous-v2` classic control benchmark. The results, presented in a new **Appendix F (Pages 25-26)**, strongly corroborate our original findings, showing that VIQS is the only method to consistently surpass its suboptimal teacher in this distinct domain.
    *   **Broader Baselines (HIL-SERL):** We have integrated **HIL-SERL**, a state-of-the-art method from the robotics domain, into our MetaDrive experiments to provide a wider comparative context (**Table 1 (Page 8)** and **Appendix G (Pages 26-27)**).

2.  **Deeper Analysis and Justification:**
    *   **Discriminator & Q-Value Stability:** We have added new analyses to empirically validate the stability of our learning framework. **Appendix D.1 (Figure 11, Pages 22-23)** now visualizes the rapid convergence and stability of the discriminator, while **Appendix E (Figure 13, Page 25)** demonstrates the stable convergence of the agent's Q-values.
    *   **Hyperparameter Robustness:** A comprehensive sensitivity analysis for our key hyperparameter, `τ_Q`, has been added in **Appendix D.2 (Figure 12, Pages 23-24)**, demonstrating the robustness of VIQS and providing a principled heuristic for its tuning.

3.  **Enhanced Clarity and Discussion:**
    *   **Bridging to Real-World Application:** We have significantly expanded the **"Limitations and Future Work" section (Page 10)** to explicitly discuss the assumption of an expert $Q$-function and provide a clear roadmap for applying VIQS with real human experts.
    *   **Clarification of Core Mechanisms:** We have refined our explanations for core concepts, such as the rationale for decoupling the trigger and shaper components, freezing the discriminator, and the value propagation mechanism, throughout the paper.

We are confident that these revisions have made our paper stronger, clearer, and more robust. We thank you once again for your constructive guidance.

---

### Comment · Area_Chair_Pkuy · 2025-11-22

Dear Reviewers,

The authors have responded to your reviews. Please review and respond to their comments.

Best,
Your AC

---

### Author Response · Authors · 2025-11-29
**Addressing Key Reviewer Concerns: A Deep Dive into Our Revisions and Their Impact**

We have synthesized the reviewers' most significant concerns into three key themes. Below, we detail our responses, the concrete new results we produced, and their direct impact on strengthening the contributions of our paper.

---

> ### Author Response · Authors · 2025-11-29
> **Concern 1: Expanding Validation to New Domains and Against Stronger Baselines**
>
> *   **Voiced by:** Reviewer fVz9, Reviewer sst8
> *   **Our Response & Impact:** We addressed this concern with **two major experimental expansions**, which not only answered the reviewers' questions but also substantially sharpened our core thesis.
>
>     1.  **Impactful Outcome 1: Proving Cross-Domain Generalizability.** We intentionally selected the `LunarLanderContinuous-v2` classic control benchmark, a domain with fundamentally different dynamics from autonomous driving. Our new experiments in **Appendix F (Page 25)** provide conclusive evidence:
>         *   **Consistent Core Finding:** VIQS is the **only method** to significantly **surpass its suboptimal teacher's performance (+16%)** under its guidance.
>         *   **Highlighting Baseline Limitations:** The competitive baseline, PVP4REAL, was "shackled" to the teacher's performance level, unable to break the ceiling, while PVP catastrophically failed. This powerfully illustrates the inherent limitations of existing methods.
>         *   **Unmatched Efficiency:** VIQS achieved this superior performance while requiring **less than half the human interventions** of PVP4REAL.
>         This new experiment is not just "one more result"; it fundamentally proves that the core principle of VIQS—"critical learning"—is **generalizable and not confined to a specific domain**.
>
>     2.  **Impactful Outcome 2: Validating Our Problem's Uniqueness and Our Method's Necessity.** We integrated HIL-SERL, a state-of-the-art method from robotics, as suggested. Critically, HIL-SERL **completely failed in our suboptimal expert setting**, achieving a 0% success rate across all expert tiers.
>         *   **This is not a critique of HIL-SERL, but one of our strongest arguments.** As detailed in our new **Appendix G**, HIL-SERL's design philosophy is **fundamentally predicated on near-optimal expert guidance**. Its failure empirically proves that **a critical gap exists** in the literature: existing high-efficiency HIL methods are brittle when the expert is imperfect.
>         *   This finding directly answers the "Why do we need VIQS?" question. It demonstrates that VIQS is not an incremental improvement but a robust solution designed for a **new and important problem quadrant** that current SOTA methods cannot handle.

---

> ### Author Response · Authors · 2025-11-29
> **Concern 2: Clarifying and Validating the Stability of Core Mechanisms**
>
> *  **Voiced by:** Reviewer fVz9, sst8, L4Uj, geqJ
> *   **Our Response & Impact:** We conducted **three new analyses** that transform VIQS from a "black box that seems to work" into a "white box with clear, predictable mechanics."
>
>     1.  **Impactful Outcome 1: Proving the Efficacy of a Stable "Quality Assessor."** Reviewers rightly questioned the stability of our discriminator, which is trained on sparse data. Our new analysis in **Appendix D.1 (Page 22)** provides a definitive answer: the discriminator is an **extremely efficient classifier**. It rapidly converges to >95% accuracy with a surprisingly small number of interventions (often <5k steps), providing strong empirical justification for our design choice to freeze it as a stable quality signal. This proves both the **feasibility and high sample efficiency** of our approach.
>
>     2.  **Impactful Outcome 2: Demonstrating Robustness and Providing Principled Hyperparameter Tuning.** Addressing concerns about the sensitivity of our key hyperparameter $τ_Q$, our new study in **Appendix D.2 (Page 23)** yielded two critical findings:
>         *   **Robust Performance:** VIQS consistently outperforms the expert across a wide range of $τ_Q$ values, proving it **does not require meticulous tuning**.
>         *   **Principled Heuristic:** More importantly, we discovered that the optimal $τ_Q$ value corresponds directly to the **initial Q-value gap** between the agent's and expert's actions. This provides a **principled, data-driven heuristic for setting $τ_Q$**, transforming it from a "magic number" into a reproducible parameter and significantly enhancing the usability of our method.
>
>     3.  **Impactful Outcome 3: Providing Direct Evidence of Stable Value Convergence.** To allay fears of value explosion in our reward-free setting, our new **Appendix E (Page 24)** visualizes the Q-values over time. The results clearly show that across all expert levels, the Q-values **exhibit stable convergence and remain bounded**, providing the final and most direct empirical proof of our framework's stability.

---

> ### Author Response · Authors · 2025-11-29
> **Concern 3: Positioning the Scientific Contribution and Bridging to Practical Application**
>
> *  **Voiced by:** Reviewer fVz9, Reviewer geqJ
> *   **Our Response & Impact:** We addressed this central concern with a strategy of **transparently acknowledging the limitation, precisely positioning our contribution, and providing a clear roadmap for future work.**
>
>     1.  **Impactful Outcome 1: Clearly Positioning the Scientific Contribution.** In the revised "Limitations" section (**Page 10**) and our responses, we explicitly frame our core goal: to prove, in a **scientifically controlled and reproducible setting**, that it is **possible to systematically break the teacher-quality ceiling by design**. We argue that the "agent-as-expert" paradigm is a **necessary scientific tool** to achieve this foundational proof, not a flaw to be hidden. This reframes the discussion from "Is it immediately deployable?" to "Does it establish a valuable scientific principle?".
>
>     2.  **Impactful Outcome 2: Providing a Clear and Feasible Bridge from Theory to Practice.** Crucially, we did not stop at acknowledging the limitation. In the "Future Work" section (**Page 10**), we leverage VIQS's inherent modularity (decoupling of trigger and learner) to propose **two concrete and viable pathways for real-world deployment**:
>         *   **Path 1 (Direct Replacement):** The $Q^E$ trigger can be replaced with **any $Q^E$-free signal**, such as a human-pressed intervention button or an uncertainty-based metric. Our core quality-aware learning module would still function critically on the resulting interventions.
>         *   **Path 2 (Learning a Proxy):** Inspired by advances like RLHF, one could **learn a proxy $Q^E$ function offline** from a pre-collected dataset of human preferences (e.g., pairwise comparisons).
>     Proposing these paths transforms our method from an idealized theoretical model into a framework with **tangible engineering potential and a clear trajectory for future research**. It demonstrates that our work is a solid foundation upon which more practical systems can be built.

---

### Meta-Review · Program_Chairs · 2026-01-06

**Summary:**

The reviewers raised the following concerns:
- Strong assumption: it is unclear if an expert Q-function  can accurately judge the agent's new actions.
- Lack deep analysis: lack clarifying and validating the stability of core mechanisms.
- Limited empirical justification: all experiments are conducted on the MetaDrive benchmark. It should be evaluated on other benchmarks to strengthen the empirical evidence.
- Lack of sensitivity analysis for key parameters.
- Conflict experimental results with the description of previous works.
- Missing details: the paper is missing clear definitions for several terms, such as in line 241 and in Eq.(6). The explanation and justification for several key mechanisms (GAN learning and Q value) are vague and inadequate.
- Gap to real-world application

**Reviewer Concerns:**

The following reviewer concerns have been partially addressed by the rebuttal:
- Limited empirical justification: added new experiments.
- Lack deep analysis: clarifed and validated the stability of core mechanisms.
- Position the Scientific Contribution
- Missing details

The concerns that are outstanding:
- Strong assumption
- Gap to real-world application

**Reviewer Scores:**

Reviewer fVz9 raised four concerns, and the rebuttal partially addressed most of them with two outstanding. Reviewer fVz9 might keep the score.

Reviewer sst8 raised three concerns, and the rebuttal addressed most of them. Reviewer sst8 might increase the score to 6.

Reviewer L4Uj raised two concerns, and the rebuttal addressed most of them. L4Uj keeps the slightly positive score.

Reviewer geqJ raised four concerns, and the rebuttal addressed three of them. geqJ keeps the slightly positive score.

---

### Decision · Program_Chairs · 2026-01-26

Reject